# DETECTING ADVERSARIAL EXAMPLES VIA NEURAL FINGERPRINTING

## ABSTRACT

Deep neural networks are vulnerable to adversarial examples: input data that has been manipulated to cause dramatic model output errors. To defend against such attacks, we propose *NeuralFingerprinting*: a simple, yet effective method to detect adversarial examples that verifies whether model behavior is consistent with a set of *fingerprints*. These fingerprints are encoded into the model response during training and are inspired by the use of biometric and cryptographic signatures. In contrast to previous defenses, our method does not rely on knowledge of the adversary and can scale to large networks and input data. In this work, we 1) theoretically analyze *NeuralFingerprinting* for linear models and 2) show that *NeuralFingerprinting* significantly improves on state-of-the-art detection mechanisms for deep neural networks, by detecting the strongest known adversarial attacks with *98-100%* AUC-ROC scores on the MNIST, CIFAR-10 and MiniImagenet (20 classes) datasets. In particular, we consider several threat models, including the most conservative one in which the attacker has full knowledge of the defender's strategy. In all settings, the detection accuracy of *NeuralFingerprinting* generalizes well to unseen test-data and is robust over a wide range of hyperparameters.

## 1 INTRODUCTION

Deep neural networks (DNNs) are highly effective pattern-recognition models for a wide range of tasks, e.g., computer vision (He et al., 2015), speech recognition (Xiong et al., 2016) and sequential decision-making (Silver et al., 2016). However, DNNs are vulnerable to adversarial examples: an attacker can add (small) perturbations to input data that are imperceptible to humans, and can drastically change the model's output, introducing catastrophic errors(Szegedy et al., 2013; Goodfellow et al., 2014). A key challenge then is to make DNNs robust to adversarial attacks, so DNNs can be reliably used in noisy environments or mission-critical applications, e.g., in autonomous vehicles. There are two broad classes of solution approaches: making robust predictions and detecting adversarial examples. A major bottleneck with robust predictions can be the inherently larger sample complexity associated with robust learning (Schmidt et al., 2018). Hence, we focus on the latter and propose *NeuralFingerprinting* (*NeuralFP*): a fast, secure and effective method to *detect* adversarial examples.

The core idea of *NeuralFP* is to encode fingerprint patterns into the response of a neural network around real (i.e., non-adversarial) data. These patterns characterize the model's expected behavior around real data and can thus be used to detect adversarial examples, around which the model outputs are not consistent with the expected fingerprint outputs (Figure 1). This approach is attractive as encoding fingerprints is feasible and simple to implement during training, and evaluating fingerprints is computationally inexpensive. Furthermore, *NeuralFP* is agnostic of the adversary's attack mechanism, and differs from recent methods (Meng & Chen, 2017; Ma et al., 2018; Lee et al., 2018) that rely on auxiliary classifiers for detecting adversarial examples.

In this work, we extensively analyze and evaluate *NeuralFP* under various settings and threat models. We theoretically analyze the feasibility and effectiveness of *NeuralFP* for linear models. Furthermore, for DNNs, we experimentally validate under multiple (including the most conservative) threat models that 1) *NeuralFP* achieves almost perfect detection AUC-ROC scores against state-of-the-art adversarial attacks on various datasets and 2) adaptive attackers with knowledge of the fingerprints fail to craft successful attacks. To summarize, our key contributions are:

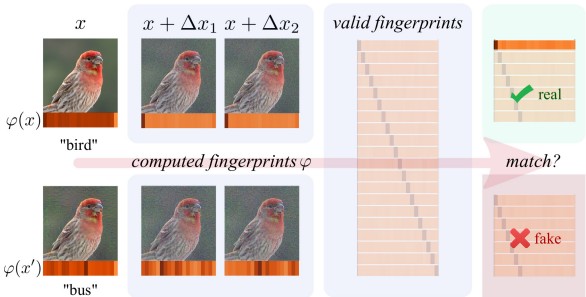

Figure 1: Detecting adversarial examples using *NeuralFP* with $N = 2$ fingerprints, for $K$-class classification. $\varphi(x)$ is the model output. *NeuralFP* separates real data $x$ (top) from adversarial data $x' = x + \eta$ (bottom) by comparing the sensitivity of the model to certain predefined perturbations around unseen inputs with a reference sensitivity encoded around the manifold of *real* images during training.

- We present *NeuralFP*: a simple and secure method to detect adversarial examples that does not rely on knowledge of the attack mechanism.

- We formally characterize the effectiveness of *NeuralFP* for linear classification.

- We empirically show on vision tasks that *NeuralFP* achieves state-of-the-art near-perfect AUC-ROC scores on detecting and separating unseen test data and the strongest known adversarial attacks.

- We empirically show that the performance of *NeuralFP* is robust to the choice of fingerprints and is effective for a wide range of choices of hyperparameters.

- Finally, we show that *NeuralFP* can be robust to known adaptive-whitebox-attacks, where an attacker has full knowledge of the fingerprint data[1].

## 1.1 RELATED WORK

Several forms of defense to adversarial examples have been proposed, primarily in the setting of *robust prediction*, including adversarial training, detection and reconstructing images using adversarial networks (Meng & Chen, 2017). Instead, our focus is on *detecting* adversarial attacks.

**Robust Prediction.** Raghunathan et al. (2018); Kolter & Wong (2017) train on convex relaxations of the network to maximize robustness and are able to formally certify robustness for small perturbations. These methods do not yet scale to large models or high-resolution inputs. Several other defenses attempt to make robust predictions: by relying on randomization (Xie et al., 2018), introducing non-linearity that is not differentiable (Buckman et al., 2018) and by relying on Generative Adversarial Networks (Song et al., 2018; Pouya Samangouei, 2018) for denoising images. However, recent work has shown that several of these defenses – including (Buckman et al., 2018; Meng & Chen, 2017; Song et al., 2018), amongst others – are not secure, when subject to a stronger adversary (Athalye et al., 2018; Uesato et al., 2018; Carlini & Wagner, 2017a;b). The reliance of these defenses on obfuscated gradients and obscurity is used to render the defenses inviable. Madry et al. (2017) employs robust-optimization techniques to minimize the maximal loss that can be achieved through first-order attacks, and has been shown to not cause obfuscated gradients (Athalye et al., 2018).

**Robust Detection.** Amongst defenses that study the detection of adversarial examples, Ma et al. (2018) detects adversarial samples using an auxiliary classifier trained to use an expansion-based measure, *local intrinsic dimensionality (LID)*. Similar detection methods based on Kernel Density (KD), Bayesian-Uncertainty (BU) (Feinman et al., 2017) and the Mahalonobis Distance (MD) (Lee et al., 2018) using artifacts from trained networks have been considered. (Carlini & Wagner, 2017a; Athalye et al., 2018) showed LID, KD and BU to be ineffective against stronger attacks. In contrast, *NeuralFP* does not depend on auxiliary classifiers and performs significantly better than LID, KD and BU: 1) when the attack mechanism is unknown and 2) is robust to stronger attacks that break LID, KD and BU. A recent defense, (Lee et al., 2018), remains to be extensively studied under a stronger threat model and for signs of gradient obfuscation and security through obscurity (Athalye et al., 2018; Uesato et al., 2018), where the dynamic attacker is aware of the defense mechanism.[2]

---

[1]Code to reproduce experiments and the model weights: `https://www.dropbox.com/sh/iq0yub74gquz1od/AADZXUVabMvZasPrt-6-c-Wpa?dl=0`.

[2] Due to unavailable implementation code, we were not able to baseline against this defense.

Further, all of the above mentioned defenses (including MD) rely on auxiliary models for detection, which often adds to the vulnerability of the defense.

**Adversarial Examples in Other Domains.** We evaluate *NeuralFP* on computer vision tasks, a domain in which DNNs have proved to be effective and adversarial examples have been extensively studied. *NeuralFP* could potentially be employed to secure DNNs against attacks in domains such as speech recognition (Carlini & Wagner, 2018) and text comprehension (Jia & Liang, 2017). Adversarial attacks have also been studied in domains such as detection of malware (Grosse et al., 2017), spam (Nelson et al., 2008) and intrusions (Wagner & Soto, 2002). Data-poisoning (Alfeld et al., 2016) is another form of attack where maliciously crafted data is injected during training.

## 2 FINGERPRINTING FOR DETECTION OF ADVERSARIAL EXAMPLES

We consider supervised classification, where we aim to learn a model $f(x;\theta)$ from *real* data $\left\{(x^i, y^{*i})\right\}_{i \in \mathcal{I}}$, where $x \in \mathbb{R}^l$ is an input example (e.g., an image) and $y^*$ is a 1-hot label vector $y^* \in \{0,1\}^K$ over $K$ classes. Here, we assume the data is sampled from a data distribution $P_{data}(x, y)$. For example, a neural network model $f$ predicts class probabilities $P(y|x;\theta)$ as:

$$f(x;\theta)_j = P(y_j|x;\theta) = \frac{\exp h(x;\theta)_j}{\sum_l \exp h(x;\theta)_l}, \tag{1}$$

where $h(x;\theta) \in \mathbb{R}^K$ are called logits and the most likely class is chosen. The optimal $\theta^*$ can be learned by minimizing a loss function $L(x, y; \theta)$, e.g., cross-entropy loss. In this setting, an attacker attempts to construct adversarial examples $x'$, such that $\hat{y} = \operatorname{argmax}_l P(y_l|x';\theta)$ is an *incorrect* class prediction (i.e., $P_{data}(x', \hat{y}) = 0$). In this work, we focus on *bounded* adversarial attacks, which produce small perturbations $\eta$ that cause mis-classification. This is a standard threat model, for an extensive review see (Akhtar & Mian, 2018). More generally, a bounded adversarial example causes a large change in model output, i.e. for $\delta, \rho > 0$:

$$\|\eta\| \leq \delta, \quad \|f(x+\eta) - f(x)\| > \rho,$$

such that the class predicted by the model changes: $\operatorname*{argmax}_j f(x+\eta)_j \neq \operatorname*{argmax}_j f(x)_j$. An example of a bounded attack is the Fast-Gradient Sign Method (FGSM) Goodfellow et al. (2014) which uses an input-space gradient: $\eta \propto \operatorname{sign} \frac{\partial L(x,y;\theta)}{\partial x}$. Since the perturbation $\eta$ is bounded and small, if $x$ is an image, $x'$ can be indistinguishable from $x$ but still cause very different predictions. Hence, our goal in this work is to efficiently detect and separate real from adversarial examples.

**Neural Fingerprinting.** To defend DNNs against adversarial attacks, we propose *NeuralFP*: a method that detects whether an input example $x$ is real or adversarial. This algorithm is summarized in Algorithm 1 and Figure 1. The key idea of *NeuralFP* is to detect adversarial examples by using a form of *consistency check of the model output around the input*. More precisely, a defender using *NeuralFP chooses* (a set of) input perturbation(s) $\Delta x$ around $x$ and checks whether the model output on $x + \Delta x$ changes according to a *chosen* $\Delta y$. These chosen output-changes are encoded into the network during training. We will discuss strategies for choosing the fingerprints $(\Delta x, \Delta y)$ in Section 2.2.

---

**Algorithm 1** *NeuralFP*

1: **Input**: example $x$, comparison function $D$ (see Eqn 3), threshold $\tau > 0$, $\chi^{i,j}$ (see Eqn 2), model $f$.
2: **Output**: `accept` / `reject`.
3: **if** $\exists j : D(x, f, \chi^{i,j}) \leq \tau$ **then**
4:     **Return:** `accept` # *x is real*
5: **else**
6:     **Return**: `reject` # *x is not real*
7: **end if**

---

Formally, we define a fingerprint $\chi$ as the tuple $\chi \triangleq (\Delta x, \Delta y)$. For $K$-class classification, we define a set of $N$ fingerprints:

$$\chi^{i,j} = (\Delta x^i, \Delta y^{i,j}), \ \ i = 1, \ldots, N, \ \ j = 1, \ldots, K, \tag{2}$$

where $\chi^{i,j}$ is the $i$th fingerprint for class $j$. As noted before, the $\Delta x^i$ ($\Delta y^{i,j}$) are chosen by the defender. Note that we use the same directions $\Delta x^i$ for each class $j = 1, 2 \ldots, K$, and that $\Delta y^{i,j}$ can be either discrete or continuous depending on $f(x;\theta)$.

To characterize sensitivity, we define the function $F(x, \Delta x^i)$ to measure the change in model output. A simple choice could be $F(x, \Delta x^i) = f(x + \Delta x^i) - f(x)$ (although we will use variations hereafter). To compare $F(x, \Delta x^i)$ with the reference output-perturbation $\Delta y^i$, we use a comparison function $D$:

$$D(x, f, \chi^{\cdot,j}) \triangleq \frac{1}{N} \sum_{i=1}^{N} \|F(x, \Delta x^i) - \Delta y^{i,j}\|_2. \tag{3}$$

Hence, the goal of a defender is to minimize $D(x, f, \chi^{i,j})$ for real data $x$.

**Detecting Adversarial Examples.** *NeuralFP* classifies a new input $x'$ as *real* if the change in model output is close to the $\Delta y^{i,j}$ *for some class* $j$, for all $i$. Here, we use a comparison function $D$ and threshold $\tau > 0$ to define the level of agreement required, i.e., we declare $x'$ *real* when $D$ is below a threshold $\tau$.

$$x' \text{ is real} \Leftrightarrow \exists j : D(x', f, \chi^{\cdot,j}) \leq \tau. \tag{4}$$

Hence, the *NeuralFP* test is defined by the data: $\texttt{NFP} = \left( \{\chi^{i,j}\}_{i=1\dots N, j=1\dots K}, D, \tau \right)$.

**Encoding Fingerprints.** Once a defender has constructed a set of desired fingerprints $\chi$, the chosen fingerprints can be embedded into the network's response by adding a fingerprint regression loss during training. Given a classification model, the fingerprint loss is:

$$L_{\text{fp}}(x, y, \chi; \theta) = \sum_{i=1}^{N} \|F(x, \Delta x^i) - \Delta y^{i,k}\|_2^2, \tag{5}$$

where $k$ is the ground truth class for example $x$ and $\Delta y^{i,k}$ are the fingerprint outputs. Note that we *only train on the fingerprints for the ground truth class*. The total training objective then is:

$$\min_\theta \sum_{(x,y)} \left( L_0(x, y; \theta) + \alpha L_{\text{fp}}(x, y, \chi; \theta) \right),$$

where $L_0$ is a loss function for the task (e.g. cross-entropy loss for classification) and $\alpha$ a positive scalar. In practice, we choose $\alpha$ such that it balances the task and fingerprint losses. The procedure trains the model so that the function $D$ has low values around the real data, e.g., the train-set. We then exploit this characterization to detect adversarial test-set examples using $D$.

**Complexity.** Extra computation comes from Algorithm 1 which requires $O(NK)$ forward passes to compute the differences $F(x, \Delta x^i)$. A straightforward implementation is to check (4) iteratively for all classes, and stop whenever an agreement is seen or all classes have been exhausted. However, this can be parallelized and performed in minibatches for real-time applications.

## 2.1 THREAT MODEL ANALYSIS

We study *NeuralFP* under various threat models (Table 1), where the attacker has varying levels of knowledge of $\texttt{NFP}$ and model $f(x; \theta)$.

In the **partial-whitebox-attack** (PWA) setting, the attacker has access to $\theta$, can query $f(x; \theta)$ and its derivatives, *but does not know NFP*. We evaluate under this setting in Section 3. This setting is the most commonly studied setting, and most defenses reported in

| $\theta$ | NFP | |
|---|---|---|
| $\times$ | $\times$ | blackbox-attack |
| $\checkmark$ | $\times$ | partial-whitebox-attack |
| $\checkmark$ | $\checkmark$ | adaptive-whitebox-attack |

Table 1: Threat models: attacker knows $\theta$ and / or NFP.

Section 1, including (Ma et al., 2018; Song et al., 2018; Lee et al., 2018), study attacks under this threat model. The PWA setting is relevant, for instance, when the attacker has a copy of the model, but the fingerprints are kept private, in accordance with Kerchhoff's principle (Shannon, 1949) (e.g. when the *NeuralFP* test is run on a cloud). Although PWA is not the strongest threat scenario, we study it because of its widespread study in recent literature (Ma et al., 2018; Song et al., 2018) and it provides a direct comparison of *NeuralFP* against established baselines.

Reverse engineering the $\texttt{NFP}$ by brute-force search can be combinatorially hard. To see this, consider a simple setting where only the $\Delta y^{i,j}$ are unknown, and that the attacker knows that each fingerprint

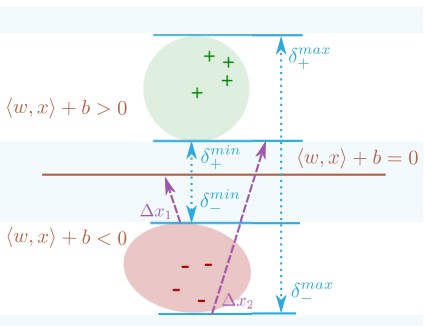

Figure 2: Geometry of fingerprints for SVMs with linearly separable data. Let $d(x)$ be the distance of $x$ to the decision boundary (see Thm 1). $\delta_\pm^{max}$ ($\delta_\pm^{min}$) denote the maximal (minimal) distances of the positive ($x_+$) and negative ($x_-$) examples to the separating hyperplane $\langle w, x \rangle + b = 0$. The fingerprint $\Delta x^1$ with $\langle \Delta x^1, \hat{w} \rangle = \delta_-^{min}$ will have $f(x_- + \Delta x) < 0$ and $f(x_-) < 0$ for all $x_-$ in the data distribution (red region). Hence, $\Delta x^1$ will flag all $x'$ in the regions $-\delta_-^{min} < d(x') < 0$ as not real ($d(x')$ is the signed distance of $x'$ from the boundary), since for those $x'$ it will *always* see a change in predicted class. Similarly, $\Delta x^2$ with $\langle \Delta x^2, \hat{w} \rangle = \delta_-^{max}$ always sees a class change for real $x_-$, flagging all $x' : d(x') < -\delta_-^{max}$ as not real.

is discrete, i.e. each component $\Delta y_k^{i,j} = \pm 1$. Then the attacker would have to search over combinatorially ($\mathcal{O}(2^{NK})$) many $\Delta y$ to find the subset of $\Delta y$ that satisfies the detection criterion in equation (4). Further, smaller $\tau$s reduce the volume of inputs accepted as real.

However, the *strongest* threat model assumes the attacker has full information about the defender, for e.g., when the attacker can either reverse-engineer or has access to the fingerprint data NFP, so that stronger attacks could be possible. Hence, we study this scenario extensively as well.

In the **adaptive-whitebox-attack** (AWA) setting, the adversary has perfect knowledge of NFP in addition to the information available under PWA. For this setting, first, in Section 2.2 we analyze *NeuralFP* for binary classification with linear models (SVMs), by characterizing the region of inputs that will be judged as real for a given set of fingerprints $\chi^{i,j}$ and the correspondence with the data distribution $P_{data}(x, y)$ (see Theorem 1). Second, we empirically show the robustness of *NeuralFP* for DNNs to adaptive attacks in Section 3.2. Ma et al. (2018); Xu et al. (2018) are other recent defenses that investigate the robustness to such adaptive attacks.

The **blackbox-attack setting** is the weakest setting, where the adversary has no knowledge of the model parameters or NFP. We evaluate *NeuralFP* under this setting in Appendix F. Additionally, we define the notion of **whitebox-defense** (the defender is aware of the attack mechanism) and **blackbox-defense** (defender has no knowledge about attacker). As such, *NeuralFP* is a blackbox-defense. There has been considerable progress in the blackbox-attack and whitebox-defense settings (e.g. (Liao et al., 2017)). However, progress in the blackbox-defense threat model is relatively limited.

## 2.2 CHOOSING AND CHARACTERIZING FINGERPRINTS: LINEAR MODELS

We first analyze *NeuralFP* on binary classification with data $\left\{ \left( x^i, y^{*i} \right) \right\}_{i \in \mathcal{I}}$ and linear model (SVM):

$$f(x) = \langle w, x \rangle + b, \quad \hat{y} = \text{sign } f(x) \in \{-1, 1\},$$

on inputs $x^i \in \mathbb{R}^n$, where $n \gg 1$ (e.g., $n = 900$ for MNIST). The SVM defines a hyperplane $f(x) = 0$ to separate positive ($\hat{y} = +1$) from negative examples ($\hat{y} = -1$). We will assume that the positive and negative examples are linearly separable by a hyperplane defined by a normal $\hat{w} = \frac{w}{\|w\|}$. We define the minimal and maximal distance from the examples to the hyperplane along $\hat{w}$ as:

$$\delta_\pm^{min} = \min_{i:y^i = \pm 1} \left| \langle x^i, \hat{w} \rangle \right|, \ \delta_\pm^{max} = \max_{i:y^i = \pm 1} \left| \langle x^i, \hat{w} \rangle \right|.$$

In this setting, the set of $x^i$ classified as real by fingerprints is determined by the geometry of $f(x)$. Here, for detection, we measure the *exact change in predicted class* using

$$F(x, \Delta x) = \Delta y = \text{sign} \left( \langle w, x + \Delta x \rangle + b \right) - \text{sign} \left( \langle w, x \rangle + b \right) \in \{-2, 0, 2\}.$$

**Theorem 1** (Fingerprint Detection for SVM). *Consider an SVM with $\hat{w} = \frac{w}{\|w\|}$ and separable data, and the following criteria:*

$$(\Delta x^1 = \delta_-^{min} \hat{w}, \Delta y^{1,-} = 0), \qquad (6) \qquad (\Delta x^3 = -\delta_+^{max} \hat{w}, \Delta y^{3,+} = -2), \qquad (8)$$

$$(\Delta x^2 = \delta_-^{max} \hat{w}, \Delta y^{2,-} = +2), \qquad (7) \qquad (\Delta x^4 = -\delta_+^{min} \hat{w}, \Delta y^{4,+} = 0). \qquad (9)$$

*An adversarial input $x' = x_\pm + \eta$ for which one of the following holds:*

$$d(x') > \delta_+^{max}, \quad 0 < d(x') < \delta_+^{min}, \quad d(x') < -\delta_-^{max}, \quad -\delta_-^{min} < d(x') < 0, \qquad (10)$$

*will satisfy one of the above listed criteria. Here, $d(x') = \frac{\langle x', w \rangle + b}{\|w\|}$ represents the signed distance of $x'$ from the separating hyperplane.*

The proof for two fingerprints is shown in Figure 2. For the full proof, see the Appendix. Theorem 1 by itself *does not* prevent attacks *parallel* to the decision boundary. An adversary could push a negative example $x_-$ across the boundary to a region outside the data distribution ($P_{data}(x_- + \eta, y) = 0$), but within distances $\delta_+^{min}$ and $\delta_+^{max}$ of the boundary. This would still be judged as real by using fingerprints. However, such examples could still be detected by also checking the distance of $x_- + \eta$ to the nearest $x_+$ in the dataset.

## 2.3 CHOOSING AND CHARACTERIZING FINGERPRINTS: DNNs

In contrast to the linear setting, in general *NeuralFP* utilizes a *softer* notion of fingerprint matching by checking whether the model outputs match *changes in normalized-logits*. Specifically, for classification models $f(x; \theta)$ with logits $h(x; \theta)$ (see Eqn 1), where $F$ is defined as:

$$F(x, \Delta x^i) \triangleq \varphi(x + \Delta x^i) - \varphi(x), \quad \varphi(x) \triangleq \frac{h(x; \theta)}{\|h(x; \theta)\|},$$

where $\varphi$ are the normalized logits. The logits are normalized so the DNN does not fit the $\Delta y$ by making the weights arbitrarily large. Here, we use $D(x, \Delta x^i)$ as in (3). Note that here $\Delta y^{i,j} \in \mathbb{R}^K$.

**Choosing $\Delta y$**  For our experiments, we choose the $\Delta y$ so that the normalized-logit of the true class either increases or decreases along the $\Delta x^i$ (analogous to the linear case). For e.g., for a 10-class classification task, if $x$ is in class $k$ we choose $\Delta y$ of the form:

$$\Delta y_{l \neq k}^k = -\alpha, \quad \Delta y_{l=k}^k = \beta, \quad k = 1, \ldots, 10, \qquad (11)$$

where $\alpha, \beta \in \mathbb{R}^+$. We found that reasonable choices are $\alpha = 0.25$ and $\beta = 0.75$, and that the method is not sensitive to these specific choices. Further, we experimented with randomizing the signs of $\Delta y^k$ and found that *NeuralFP*'s performance is robust to this randomization as well. [3]

**Choosing $\Delta x$**  For nonlinear models (e.g., DNNs), the best fingerprint-direction choice is not obvious. To overcome this, we propose a straightforward extension from the linear case, using randomly sampled fingerprints. Randomization minimizes structural assumptions that may make *NeuralFP* exploitable. We empirically found that this provides effective detection.

For all experiments, we sampled the fingerprint directions $\Delta x^i$ from a uniform distribution ($\Delta x^i \sim \mathcal{U}(-\varepsilon, \varepsilon)^l$), where $l$ is the input dimension, and each pixel is uniformly randomly sampled in the range $[-\varepsilon, \varepsilon]$. Our experiments (see Figure 6) suggest that *NeuralFP* is *not sensitive* to the random values sampled. This also suggests that the $\Delta x^i$ could be chosen based on a different approach.

**Visualizing Fingerprints.**  To understand the behavior of fingerprinted non-linear models we trained a neural network (two hidden layers with 200 ReLU nodes each) to distinguish between two Gaussian balls in a 2D space (Figure 3, left). Without fingerprints, the model learns an almost linear boundary separating the two balls (compare with Figure 2). When we train to encode the fingerprints (negative of $\Delta y$s in (11)), we observe that *NeuralFP* causes the model to learn a highly non-linear boundary (Figure 3, center) forming pockets of low fingerprint-loss characterizing the data-distribution (Figure 3,right). In this simple setting, *NeuralFP* learns to delineate the data-distribution, where the darker regions are accepted as real and the rest is rejected (Figure 4).

---

[3] Using $\Delta y_{l \neq k}^{i,k} = -\alpha \cdot (2p - 1), \Delta y_{l=k}^{i,k} = \beta \cdot (2p - 1), p \sim \text{Bern}\left(\frac{1}{2}\right)$, *NeuralFP* achieves AUC-ROC > 95% on PWAs. Here $p \in \{0, 1\}$ is a Bernoulli random variable that is resampled for each $\Delta x^i$.

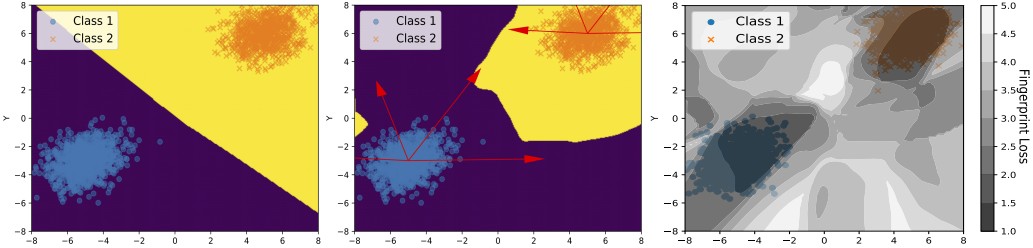

Figure 3: Left: decision boundary without fingerprints. Center: with fingerprints, red arrows indicate fingerprint-directions. The decision boundary is significantly more non-linear. Right: contour plot of fingerprint loss. *NeuralFP* detects dark regions as "real", while lighter regions are "fake" (tunable through $\tau$). Fingerprinting create valleys of low-loss delineating the data-distribution from outliers.

## 3 EVALUATING *NeuralFP* ON DETECTION OF ADVERSARIAL ATTACKS

We now empirically validate the effectiveness of *NeuralFP* on a number of vision data-sets, as well as analyze the behavior and robustness of *NeuralFP* under the various threat models. We evaluate *NeuralFP* on distinguishing between unseen real images and adversarial images, for data and models of varying scales in the three threat-models discussed earlier (PWA, AWA, BA). Further, we study the sensitivity of *NeuralFP* to varying hyperparameters. The study of the blackbox-attack setting is deferred to Appendix F. We empirically find that:

- the defense is robust under all the three threat models, and
- using *NeuralFP does not diminish test accuracy*.

### 3.1 DETECTION OF PARTIAL-WHITEBOX ATTACKS

We report the AUC-ROC of *NeuralFP* on MNIST, CIFAR-10 and MiniImagenet-20 against four state-of-the-art partial-whitebox attacks (Table 2):

- *Fast Gradient Method* (FGSM)Goodfellow et al. (2014) and *Basic Iterative Method* (BIM) Kurakin et al. (2016) are both gradient based attacks with BIM being an iterative variant of FGSM. We consider both BIM-a (iterates until misclassification has been achieved) and BIM-b (iterates 50 times).
- *Jacobian-based Saliency Map Attack* (JSMA) Papernot et al. (2015) perturbs pixels using a saliency map.
- *Carlini-Wagner Attack* (CW-$L_2$): an optimization-based attack, is one of the strongest known attacks (Carlini & Wagner, 2016; Carlini & Wagner, 2017a), and optimizes to minimize the perturbation needed for misclassification.

Following Ma et al. (2018), for each dataset we consider a randomly sampled *pre-test*-set of unseen 1328 test-set images, and discard misclassified pre-test images. For the *test-set* of remaining images, we generate adversarial perturbations by applying each of the above mentioned attacks. We report AUC-ROC on sets composed in equal parts of the *test-set* and *test-set* adversarial samples. The AUC-ROC is computed by varying the threshold $\tau$. See Appendix for model and dataset details.

The baselines are LID Ma et al. (2018), a recent detection based defense; KD; BU Feinman et al. (2017); all trained on FGSM, as in Ma et al. (2018). The FGSM, BIM-a, BIM-b and JSMA attacks are untargeted. We use published code for the attacks and code from Ma et al. (2018) for the baselines.

**MNIST.** We trained a 5-layer ConvNet to $99.2 \pm 0.1\%$ test-accuracy. The set of $\Delta x^i \in \mathbb{R}^{28 \times 28}$ is chosen at random, with each pixel perturbation chosen uniformly in $[-\varepsilon, \varepsilon]$. For each $\Delta x^i$, if $x$ is of label-class $k$, $\Delta y^{i,k} \in \mathbb{R}^{10}$ is chosen to be such that $\Delta y^{i,k}_{l \neq k} = -0.235$ and $\Delta y^{i,k}_{l=k} = 0.73$, with $\|\Delta y\|_2 = 1$. The AUC-ROCs for the best $N$ and $\varepsilon$ using grid-search are reported in Table 4. We see that *NeuralFP* achieves near-perfect detection with AUC-ROC of $99 - 100\%$ across all attacks.

| Data | Attack | Test Accuracy | Bound on Adversarial Perturbation $\eta$ |
|---|---|---|---|
| MNIST | FGSM | 11.87% | $\|\eta\|_\infty \leq 0.4$ |
| | BIM-a | 0.00% | $\|\eta\|_\infty \leq 0.4$ |
| | BIM-b | 0.00% | $\|\eta\|_\infty \leq 0.4$ |
| | JSMA | 1.73% | |
| | CW-$L_2$ | 0.00% | |
| CIFAR | FGSM | 11.39 % | $\|\eta\|_\infty \leq 0.05$ |
| | BIM-a | 0.00% | $\|\eta\|_\infty \leq 0.05$ |
| | BIM-b | 0.00% | $\|\eta\|_\infty \leq 0.05$ |
| | JSMA | 13.33% | |
| | CW-$L_2$ | 0.00% | |
| MiniImagenet | FGSM | 100% | $\|\eta\|_\infty \leq 16/255$ |
| | BIM-b | 100% | $\|\eta\|_\infty \leq 16/255$ |

Table 2: Parameters and model test-accuracy on PW-attacks for different datasets (without *NeuralFP* test). CW-$L_2$ and JSMA attacks are unbounded. The bounds are relative to images with pixel intensities in [0, 1].

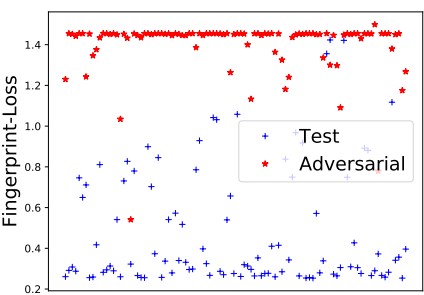

Figure 4: Fingerprint losses (mean over $N$ fingerprints) on 100 random test (blue) and adversarial (red) CIFAR-10 images. We see a clear separation in loss, illustrating that *NeuralFP* is effective across many thresholds $\tau$.

| Data | Method | FGSM | JSMA | BIM-a | BIM-b | CW-$L_2$ |
|---|---|---|---|---|---|---|
| MNIST | LID | 99.68 | 96.36 | 99.05 | 99.72 | 98.66 |
| | KD | 81.84 | 66.69 | 99.39 | 99.84 | 96.94 |
| | BU | 27.21 | 12.27 | 6.55 | 23.30 | 19.09 |
| | KD+BU | 82.93 | 47.33 | 95.98 | 99.82 | 85.68 |
| | *NeuralFP* | **100.0** | **99.97** | **99.94** | **99.98** | **99.74** |
| CIFAR-10 | LID | 82.38 | 89.93 | 82.51 | 91.61 | 93.32 |
| | KD | 62.76 | 84.54 | 69.08 | 89.66 | 90.77 |
| | BU | 71.73 | 84.95 | 82.23 | 3.26 | 89.89 |
| | KD+BU | 71.40 | 84.49 | 82.07 | 1.1 | 89.30 |
| | *NeuralFP* | **99.96** | **99.91** | **99.91** | **99.95** | **98.87** |

Table 4: Detection AUC-ROC of blackbox-defenders (do not know attack strategy) against *partial-whitebox-attackers* (know model $f(x; \theta)$, but not defense details; see Section 2.1), on MNIST, CIFAR-10 on test-set ("real") and corresponding adversarial ("fake") samples (1328 *pre-test* samples each). *NeuralFP* outperforms baselines (LID, KD, BU) on MNIST & CIFAR-10 across attacks.

**CIFAR-10.** For CIFAR-10, we trained a 7-layer ConvNet (similar to (Carlini & Wagner, 2016)) to $85 \pm 1\%$ accuracy. The $\Delta x^i$ and $\Delta y^{i,j}$ are chosen similarly as for MNIST. Across attacks, *NeuralFP* outperforms LID on average by **11.77%** and KD+BU, KD, BU even more substantially (Table 4). Even compared to LID-whitebox (where LID is aware of the attackers mechanism but *NeuralFP* is not), *NeuralFP* outperforms LID-whitebox on average by **8.0%** (Appendix, Table 10).

**MiniImagenet-20.** We also test on *MiniImagenet*-20 with 20 classes randomly chosen from the 100 MiniImagenet classes (Vinyals et al., 2016) and trained an AlexNet network on 10,600 images (not downsampled) with $91.1\%$ top-1 accuracy. We generated test-set adversarial examples using BIM-b with 50 steps (NIP) and FGSM. *NeuralFP* achieves

| Data | FGSM | BIM-b |
|---|---|---|
| MiniImagenet-20 | 99.96 | 99.68 |

Table 5: Detection AUC-ROC of *NeuralFP* vs *partial-whitebox-attacks* on MiniImagenet-20, $N = 20, \varepsilon = 0.05$.

AUC-ROCs of $> 99.5\%$ (Table 5). We could not get results for JSMA and CW-$L_2$, which require too much computation for tasks of this size. Results for other defenses are not reported due to time constraints & unavailable implementations.

**Visualization.** Figure 4 shows that fingerprint-loss differs significantly for most test and adversarial samples (across the 5 attacks in Table 2), enabling *NeuralFP* to achieve close to $100\%$ AUC-ROC.

## 3.2 DETECTION OF ADAPTIVE-WHITEBOX ATTACKS

The strongest threat-model, AWA, is one where the adversary has access to the parameters of *NeuralFP*. To evaluate whether *NeuralFP* is robust in this setting, or if it relies on gradient obfuscation or obscurity (for a detailed analysis, see Section 4), we consider adaptive variants of FGSM, BIM-b, CW-$L_2$, and SPSA (Uesato et al., 2018). Under AWA, the attacker tries to find an adversarial example

| Data | Method | Adaptive-FGSM | Adaptive-BIM-b | Adaptive-CW-$L_2$ | Adaptive-CW-$L_2$ ($\gamma_2 = 1$) | Adaptive-SPSA |
|------|--------|---------------|----------------|-------------------|-----------------------------------|---------------|
| MNIST | *NeuralFP* | 99.91 | 99.37 | 95.04 | 99.17 | 99.94 |
| CIFAR-10 | *NeuralFP* | 99.99 | 99.92 | 97.19 | 97.56 | 99.99 |

Table 6: Detection AUC-ROC for adaptive attacks on datasets MNIST and CIFAR-10. Other defenses such as (Song et al., 2018; Liao et al., 2017), including the baselines KD and BU, fail under adaptive-attacks ($< 10\%$ accuracy). For MNIST, the *NeuralFP* parameters for FGSM, SPSA are $(\varepsilon, N) = (0.1, 10)$ and $(\varepsilon, N) = (0.05, 20)$ for the BIM-b, CW-$L_2$ attacks. For CIFAR-10, the parameters are set at $(\varepsilon, N) = (0.003, 30)$ across attacks.

$x'$ that also minimizes the fingerprint-loss (5), while attacking the model trained with *NeuralFP*. We find that *NeuralFP* is robust across all AW-attacks, achieving AUC-ROCs of 96-100% (See Table 6)

**Adaptive-FGSM, Adaptive-BIM-b, Adaptive-SPSA**    For the FGSM, BIM-b and SPSA (untargeted) attacks we mount an adaptive attack with a modified optimization objective as in (Uesato et al., 2018). Specifically, for SPSA, the loss function to minimize is:

$$J_\theta^{\mathrm{adv}}(x', y^*, \theta) + \gamma L_{\mathrm{fp}}(x', y^*, \chi; \theta),$$

where $J_\theta^{\mathrm{adv}}$ is the original adversarial objective from (Uesato et al., 2018). For the gradient-based FGSM and BIM-B attacks, we use gradients of the following loss function:

$$L_{CE}(x, y^*, \theta) - \gamma L_{\mathrm{fp}}(x, y^*, \chi; \theta),$$

where $L_{CE}(x, y^*)$ is the cross-entropy loss. For each of the attacks and for each data-point, we choose the largest $\gamma \in [10^{-3}, 10^4]$ that results in a successful attack with a bisection search over $\gamma$ – note that larger $\gamma$ values increase the priority for minimizing $L_{fp}$. For the three adaptive attacks, the perturbation bounds are $\|\eta\|_\infty \leq 0.4$ for MNIST and $\|\eta\|_\infty \leq 0.05$ for CIFAR-10.

**Adaptive-CW-$L_2$**    We consider two adaptive variants of the CW-$L_2$ attack. The first variant we consider is with the modified objective function:

$$\min_{x'} \|x - x'\|_2 + \gamma_1 (L_{\mathrm{CW}}(x') + \gamma_2 L_{\mathrm{fp}}(x', y^*, \chi; \theta)). \tag{12}$$

Here, $y^*$ is the label-vector, $\gamma_1 \in [10^{-3}, 10^6]$ and $\gamma_2 \in [10^{-3}, 10^4]$ are scalar coefficients, $L_{\mathrm{fp}}$ is the fingerprint-loss we trained on and $L_{\mathrm{CW}}$ is an objective encouraging misclassification. To find $\gamma_1$ and $\gamma_2$ we do a bisection search, first decreasing $\gamma_1$ (as in Carlini & Wagner (2017a)) and then increasing $\gamma_2$ gradually in a similar manner. Note that increasing $\gamma_2$ increases the importance given to minimizing $L_{\mathrm{fp}}$. The successful attack with the smallest $L_{\mathrm{fp}}$ during our search is chosen.

The second variant is similar to the one considered in (Carlini & Wagner, 2017a). Here $\gamma_2$ is held at 1.0 and the successful attack with the smallest $\|x - x'\|_2$ is chosen during a bisection search over $\gamma_1$.

# 4 DISCUSSION AND ANALYSIS OF EMPIRICAL RESULTS

**Robustness Across Attacks.**    In addition to being robust to blackbox attacks, we find that *NeuralFP* is robust across a set of adaptive attacks. The fingerprint loss $L_{fp}$ is differentiable, and accordingly, the gradient-free adaptive attack (SPSA) does not perform better than the gradient based ones (CW-$L_2$, BIM-b, FGSM). Other recent defenses such as (Song et al., 2018; Liao et al., 2017), including the baselines KD and BU, were shown as not viable ($< 10\%$ accuracy) under adaptive attacks similar to the ones we consider in our work (Carlini & Wagner, 2017a; Uesato et al., 2018).

Adversarial training and robust optimization based defenses (Madry et al., 2017; Kannan et al., 2018) that perform robust prediction show decaying accuracy with increasing adversarial distortion. However, *NeuralFP* shows robust performance across varying adversarial distortion (See Figure 5). A possible explanation is that *NeuralFP* functions by rejecting regions that are further away from the data-distribution (Figure 3). Empirically, *NeuralFP* is less sensitive to small perturbations and more robust to larger distortions, and this is shown by the degradation in performance against the adaptive BIM & CW-$L_2$ attacks (which produce significantly smaller distortions relative to SPSA & FGSM).

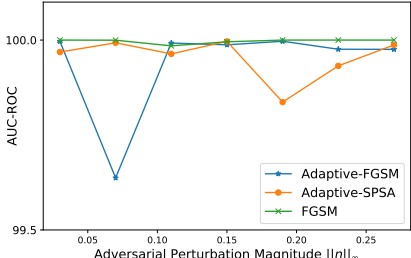

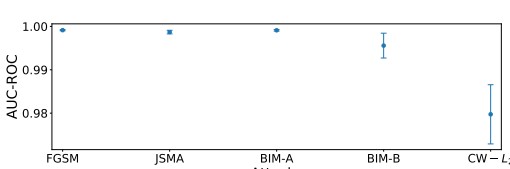

Figure 5: AUC-ROC with varying adversarial distortion (300 pre-test images). Unlike robust prediction(Madry et al., 2017; Kannan et al., 2018), *NeuralFP* shows no performance decline with increasing distortion (AUC-ROC >99%).

Figure 6: AUC-ROC mean $\mu$ and standard-deviation $\sigma$ for 32 randomly sampled finger-prints (including randomizing $N$) for CIFAR-10. The AUC-ROC across all PWAs varies little ($\sigma < 1\%$), with $\sigma$ highest for CW-$L_2$.

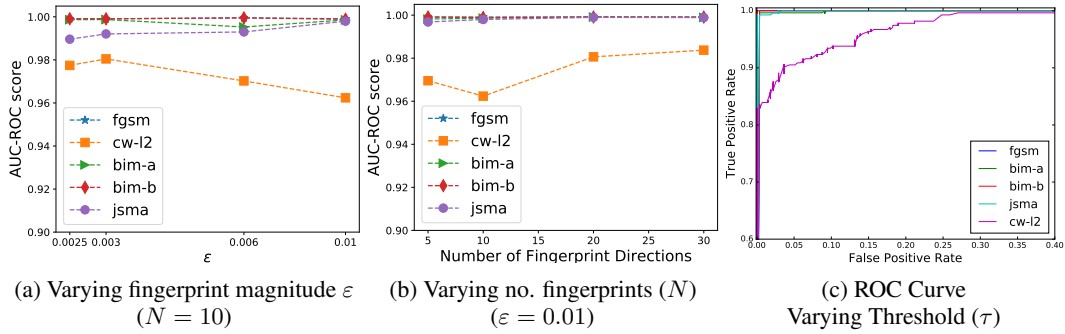

(a) Varying fingerprint magnitude $\varepsilon$     (b) Varying no. fingerprints ($N$)     (c) ROC Curve
($N = 10$)                       ($\varepsilon = 0.01$)              Varying Threshold ($\tau$)

Figure 7: AUC-ROC for different hyperparameters (left, middle) and ROC curves (right) on CIFAR-10 for partial-whitebox attacks. For analysis on MNIST, see Appendix. *NeuralFP* is robust across attacks & hyperparameters with an AUC-ROC between $95 - 100\%$. Increasing $N$ improves performance, indicating more fingerprints are harder to fool. Increasing the magnitude $\varepsilon$ decreases AUC on CW-$L_2$ only, suggesting that as adversarial perturbations become of smaller magnitude, *NeuralFP* requires $\varepsilon$.

**Obfuscating Gradients and Obscurity.** Athalye et al. (2018) argued that several previous defenses are vulnerable as they mask the true gradients ("gradient obfuscation"). A similar phenomenon was observed in Uesato et al. (2018), where such defenses were successfully attacked (i.e., <10% defense success rate) by exploiting their reliance on obscurity and masked gradients. In contrast, our experiments suggest that *NeuralFP* does not rely on either obfuscating gradients or obscurity. First, Athalye et al. (2018) argue that defenses that rely on obfuscating gradients likely perform worse on iterative attacks (e.g., BIM) compared to non-iterative ones (e.g., FGSM). However, *NeuralFP* is robust to both types of attack (Tables 4, 10). Second, *NeuralFP* is robust against gradient-free (e.g., SPSA) adaptive attacks, which suggests that it does not only rely on obscured gradients. Gradient-free attacks can be used to accurately measure robustness in the presence of obscured gradients. Finally, Carlini & Wagner (2017a); Uesato et al. (2018) argue that defenses that rely on obscurity are vulnerable to adaptive attacks. However, *NeuralFP* is robust against a variety of adaptive attacks.

**Sensitivity and Efficiency Analysis** Next, we study the effect of changing $N$ (number of finger-print directions) and $\varepsilon$ (magnitude of fingerprint-perturbation $\Delta x$) on the AUC-ROC for CIFAR-10 (for analysis on MNIST, see the Appendix). Figure 7 and 9 show that *NeuralFP* performs well across a wide range of hyperparameters and is robust to variation in the hyperparameters for PWAs. With increasing $\varepsilon$, the AUC-ROC for CW-$L_2$ decreases. As discussed before, a possible explanation is that CW-$L_2$ produces smaller adversarial perturbations than other attacks, and for larger fingerprint-distortions $\varepsilon$, the fingerprints are less sensitive to those small adversarial perturbations. However, the degradation in performance is not substantial ($\sim 4-8\%$) as we increase $\varepsilon$ over an order of magnitude. With increasing $N$, the AUC-ROC generally increases across attacks. We conjecture that larger sets of fingerprints can detect perturbations in more directions and results in better detection.

Figure 6 shows that *NeuralFP* achieves mean AUC-ROC of $98\% - 100\%$ against all PWA, with standard deviation $< 1\%$. This suggests that *NeuralFP* is *not very sensitive to the chosen fingerprints*.

## 5 CONCLUSION

Our experiments suggest that *NeuralFP* is an effective method for detecting the strongest known state-of-the-art adversarial attacks, and the high AUC-ROC scores indicate that the fingerprints generalize well to the test-set, but not to adversarial examples. Open questions are if there are attacks that can fool *NeuralFP* or if it is provably robust to certain attacks. Learning the fingerprints during training, and studying *NeuralFP* within a robust optimization framework are other interesting directions.

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

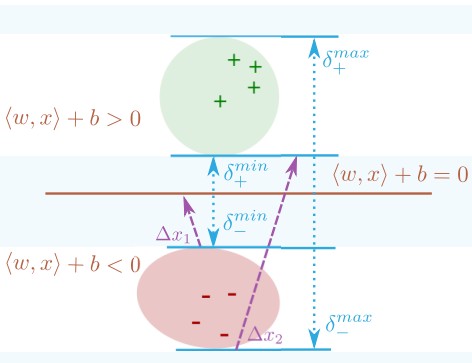

Figure 8: Geometry of fingerprints for SVMs with linearly separable data. Let $d(x)$ be the distance of $x$ to the decision boundary (see Thm 1). $\delta_{\pm}^{max}$ ($\delta_{\pm}^{min}$) denote the maximal (minimal) distances of the positive ($x_+$) and negative ($x_-$) examples to the separating hyperplane $\langle w, x \rangle + b = 0$. The fingerprint $\Delta x^1$ with $\langle \Delta x^1, e \rangle = \delta_-^{min}$ will have $f(x_- + \Delta x) < 0$ and $f(x_-) < 0$ for all $x_-$ in the data distribution (red region). Hence, $\Delta x^1$ will flag all $x'$ in the regions $-\delta_-^{min} < d(x') < 0$ as "fake", since for those $x'$ it will *always* see a change in predicted class. Similarly, $\Delta x^2$ with $\langle \Delta x^2, e \rangle = \delta_-^{max}$ always sees a class change for real $x_-$, thus flagging all $x'$ with $d(x') < -\delta_-^{max}$ as "fake".

Yang Song, Taesup Kim, Sebastian Nowozin, Stefano Ermon, and Nate Kushman. Pixeldefend: Leveraging generative models to understand and defend against adversarial examples. *International Conference on Learning Representations*, 2018. URL https://openreview.net/forum?id=rJUYGxbCW.

Christian Szegedy, Wojciech Zaremba, Ilya Sutskever, Joan Bruna, Dumitru Erhan, Ian Goodfellow, and Rob Fergus. Intriguing properties of neural networks. *arXiv preprint arXiv:1312.6199*, 2013.

Jonathan Uesato, Brendan O'Donoghue, Pushmeet Kohli, and Aaron van den Oord. Adversarial risk and the dangers of evaluating against weak attacks. In Jennifer Dy and Andreas Krause (eds.), *Proceedings of the 35th International Conference on Machine Learning*, volume 80 of *Proceedings of Machine Learning Research*, pp. 5025–5034, Stockholmsmässan, Stockholm Sweden, 10–15 Jul 2018. PMLR. URL http://proceedings.mlr.press/v80/uesato18a.html.

Oriol Vinyals, Charles Blundell, Tim Lillicrap, koray kavukcuoglu, and Daan Wierstra. Matching networks for one shot learning. In D. D. Lee, M. Sugiyama, U. V. Luxburg, I. Guyon, and R. Garnett (eds.), *Advances in Neural Information Processing Systems 29*, pp. 3630–3638. Curran Associates, Inc., 2016. URL http://papers.nips.cc/paper/6385-matching-networks-for-one-shot-learning.pdf.

David Wagner and Paolo Soto. Mimicry attacks on host-based intrusion detection systems. In *Proceedings of the 9th ACM Conference on Computer and Communications Security*, CCS '02, pp. 255–264, New York, NY, USA, 2002. ACM. ISBN 1-58113-612-9. doi: 10.1145/586110.586145. URL http://doi.acm.org/10.1145/586110.586145.

Cihang Xie, Jianyu Wang, Zhishuai Zhang, Zhou Ren, and Alan Yuille. Mitigating adversarial effects through randomization. *International Conference on Learning Representations*, 2018. URL https://openreview.net/forum?id=Sk9yuql0Z.

Wayne Xiong, Jasha Droppo, Xuedong Huang, Frank Seide, Mike Seltzer, Andreas Stolcke, Dong Yu, and Geoffrey Zweig. Achieving human parity in conversational speech recognition. *CoRR*, abs/1610.05256, 2016. URL http://arxiv.org/abs/1610.05256.

Weilin Xu, David Evans, and Yanjun Qi. Feature squeezing: Detecting adversarial examples in deep neural networks. In *Network and Distributed Systems Security Symposium (NDSS) 2018*, volume abs/1704.01155, 2018.

## APPENDIX A   PROOF FOR THEOREM 1

*Proof.* Also see Figure 8. Consider any perturbation $\eta = \lambda e$ that is positively aligned with $w$, and has $\langle \eta, e \rangle = \delta_-^{min}$. Then for any negative example $(x_-, -1)$ (except for the support vectors that lie exactly $\delta_-^{min}$ from the hyperplane), adding the perturbation $\eta$ does not change the class prediction:

$$\text{sign } f(x_-) = -1, \quad \text{sign } f(x_- - \eta) = -1. \tag{13}$$

| Layer | Parameters |
| --- | --- |
| Convolution + ReLU + BatchNorm | $11 \times 11 \times 64$ |
| MaxPool | $3 \times 3$ |
| Convolution + ReLU + BatchNorm | $5 \times 5 \times 192$ |
| MaxPool | $3 \times 3$ |
| Convolution + ReLU + BatchNorm | $3 \times 3 \times 384$ |
| MaxPool | $3 \times 3$ |
| Convolution + ReLU + BatchNorm | $3 \times 3 \times 256$ |
| MaxPool | $3 \times 3$ |
| Convolution + ReLU + BatchNorm | $3 \times 3 \times 156$ |
| MaxPool | $3 \times 3$ |
| Fully Connected + ReLU + BatchNorm | 3072 |
| Dropout | - |
| Fully Connected + ReLU + BatchNorm | 1024 |
| Dropout | - |
| Softmax | 20 |

Table 7: MiniImagenet-20 Model Used

| Layer | Parameters |
| --- | --- |
| Convolution + ReLU + BatchNorm | $5 \times 5 \times 32$ |
| MaxPool | $2 \times 2$ |
| Convolution + ReLU + BatchNorm | $5 \times 5 \times 64$ |
| MaxPool | $2 \times 2$ |
| Fully Connected + ReLU + BatchNorm | 200 |
| Fully Connected + ReLU + BatchNorm | 200 |
| Softmax | 10 |

Table 8: MNIST Model Used

The fingerprint in (6) is an example of such an $\eta$. However, if $\lambda$ is large enough, that is:

$$\langle \eta, e \rangle = \delta_-^{max}, \tag{14}$$

(e.g. the fingerprint in (7)), for *all* negative examples $(x_-, -1)$ the class prediction will *always* change (except for the $x_-$ that lie exactly $\delta_-^{max}$ from the hyperplane):

$$\text{sign } f(x_-) = -1, \quad \text{sign } f(x_- + \eta) = +1, \tag{15}$$

Note that if $\eta$ has a component smaller (or larger) than $\delta_\pm^{min}$, it will exclude *fewer* (more) examples, e.g. those that lie closer to (farther from) the hyperplane. Similar observations hold for fingerprints (8) and (9) and the positive examples $x_+$. Hence, it follows that for any $x$ that lies too close to the hyperplane (closer than $\delta_\pm^{min}$), or too far (farther than $\delta_\pm^{max}$), the model output after adding the four fingerprints will never perfectly correspond to their behavior on examples $x$ from the data distribution. For instance, for any $x$ that is closer than $\delta_+^{min}$ to the hyperplane, (9) will always cause a change in class, while none was expected. Similar observations hold for the other regions in (10). Since the SVM is translation invariant parallel to the hyperplane, the fingerprints can only distinguish examples based on their distance perpendicular to the hyperplane. Hence, this choice of $\lambda$s is optimal. □

## APPENDIX B    RANDOMIZED FINGERPRINTS

Instead of simple $\Delta y^{i,j}$, we can encode more complex fingerprints that are harder to guess for an adversary. For instance, we were able to train a network on CIFAR-10 using random $\Delta y^{i,j}$: for each $\Delta x^i$, if $x$ is in class $k$:

$$\Delta y_{l \neq k}^{i,k} = -0.235p, \Delta y_{l=k}^{i,k} = 0.7p. \tag{16}$$

| Layer | Parameters |
|---|---|
| Convolution + ReLU + BatchNorm | $3 \times 3 \times 32$ |
| Convolution + ReLU + BatchNorm | $3 \times 3 \times 64$ |
| MaxPool | $2 \times 2$ |
| Convolution + ReLU + BatchNorm | $3 \times 3 \times 128$ |
| Convolution + ReLU + BatchNorm | $3 \times 3 \times 128$ |
| MaxPool | $2 \times 2$ |
| Fully Connected + ReLU + BatchNorm | 256 |
| Fully Connected + ReLU + BatchNorm | 256 |
| Softmax | 10 |

Table 9: CIFAR Model Used

Here $p$ is a random variable with $Pr(p = 1) = 0.5$ and $Pr(p = -1) = 0.5$ that is resampled for each $\Delta x^i$, making it prohibitively hard for a brute-force attacker to guess. For this NFP, we achieve AUC-ROCs of $> 95\%$ across attacks with $N = 30, \varepsilon = 0.05$, without extensive tuning. The high model-capacity of neural networks allows to learn such complex patterns, that can be hard to reverse engineer. This also indicates that the proposed approach is robust and the specific choice of fingerprints or the distributions they are sampled from do not actually influence the detection performance to a large extent.

## APPENDIX C    MODELS FOR EVALUATION

Note: Code for CW-adaptive is based on code from https://github.com/carlini/nn_robust_attacks. Code for the other attacks was obtained from the paper (Ma et al., 2018).

### C.1    MNIST

For MNIST, we use the model described in Table 8.

### C.2    CIFAR-10

For CIFAR-10, we use the model described in Table 9

### C.3    MINIIMAGENET-20

**MiniAlexNet Model**    We use a model similar to AlexNet for MiniImagenet-20. The model used is described in Table 7

**MiniImagenet-20 classes**    We use images from the following 20 ImageNet classes for our experiments:
```
n01558993, n02795169, n03062245, n03272010, n03980874, n04515003
n02110063, n02950826, n03146219, n03400231, n04146614, n04612504,
n02443484, n02981792, n03207743, n03476684, n04443257, n07697537
```

| Data | Method | FGM | JSMA | BIM-a | BIM-b | CW-$L_2$ |
|------|--------|-----|------|-------|-------|----------|
| MNIST | LID | 99.68 | 98.67 | 99.61 | 99.90 | 99.55 |
|  | *NeuralFP* | **100.0** | **99.97** | **99.94** | **99.98** | **99.74** |
| CIFAR-10 | LID | 82.38 | 95.87 | 82.30 | 99.78 | **98.94** |
|  | *NeuralFP* | **99.96** | **99.91** | **99.91** | **99.95** | 98.87 |

Table 10: Detection AUC-ROC for *NeuralFP*,whitebox-LID against *whitebox-attackers* (know model $f(x;\theta)$, but not fingerprints; see Section 2.1), on MNIST, CIFAR-10 tasks on test-set ("real") and corresponding adversarial ("fake") samples (1328 *pre-test* samples each). *NeuralFP* outperforms the baselines (LID, KD, BU) on MNIST and CIFAR-10 across all attacks, except CW-L2 where it performs comparably. A possibly explanation for LID's improved performance against stronger, iterative attacks is gradient masking Athalye et al. (2018).

## APPENDIX D   SENSITIVITY ANALYSIS

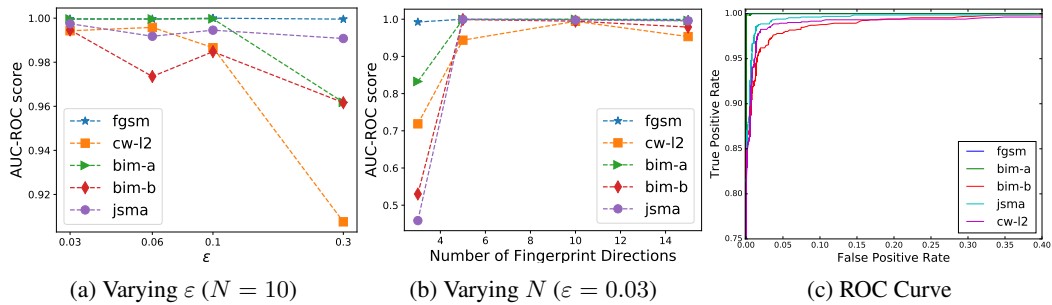

(a) Varying $\varepsilon$ ($N = 10$)        (b) Varying $N$ ($\varepsilon = 0.03$)        (c) ROC Curve

Figure 9: AUC-ROC performance for different hyperparameter settings (left, middle) and ROC curves (right) on MNIST. We see that the performance of *NeuralFP* is robust across attacks and hyperparameters, with the AUC-ROC between $90 - 100\%$ for most settings. The AUC-ROC is lowest versus CW-$L_2$, which is one of the strongest known attack.

## APPENDIX E   *NeuralFP* VS WHITEBOX-LID

We also compare LID against *NeuralFP* in the setting where LID is aware of the attack mechanism and trains against the specific attack mechanism, while *NeuralFP* still does not use any attacker information(Table 9). In this *unfair* setting, we see LID performs comparably on CW-L2, and *NeuralFP* outperforms LID against all other attacks. Athalye et al. (2018) provides a possible explanation for this behavior for LID, where defenses relying on obfuscated gradients perform better against stronger attacks (CW-$L_2$) compared to one-step weaker attacks.

## APPENDIX F   EVALUATING *NeuralFP* AGAINST BLACKBOX ATTACKS

In Athalye et al. (2018), the authors indicate that testing against blackbox attacks is useful to gauge if the defense is relying on gradient masking. To explore this aspect of our defense, we construct adversarial examples for models trained without fingerprinting and then test the ability of fingerprinted models to distinguish these from unseen test-data. We find that the performance does not degrade from the whitebox-attack setting, in contrast to other defenses evaluated in Athalye et al. (2018), where the performance degrades against blackbox-attacks.

## APPENDIX G   EVALUATED ATTACK HYPERPARAMETERS

Table 12 summarizes the hyperparameter settings corresponding to the experiments in Tables 4,6.

| Data | Method | FGM |
|---|---|---|
| MNIST | *NeuralFP* | **99.96** |
| CIFAR-10 | *NeuralFP* | **99.92** |

Table 11: Detection AUC-ROC for *NeuralFP* against *blackbox-attackers* (know dataset but not model or fingerprints), on MNIST, CIFAR-10 tasks on test-set ("real") and corresponding blackbox adversarial ("fake") samples (1328 *pre-test* samples each). *NeuralFP* achieves near perfect AUC-ROC scores in this setting against FGM. Iterative attacks are known to not transfer well in the blackbox setting. For CIFAR-10, the hyperparameters are $N = 30, \varepsilon = 0.003$ and for MNIST, the hyperparameters are $N = 10, \varepsilon = 0.03$. We did not tune the parameters because this setting in itself achieved near perfect detection rates.

| Method | Parameters |
|---|---|
| CW-$L-2$ | Bisection-steps ($\gamma_1$)=9, Bisection-steps ($\gamma_2$)=5, Max Iteration Steps = 1000, L2_ABORT_EARLY = True, L2_LEARNING_RATE = 1e-2, L2_TARGETED = True, L2_CONFIDENCE = 0, L2_INITIAL_CONST = 1e-3 ($\gamma_1$ initial value) L2_INITIAL_CONST_2 = 0.1 ($\gamma_2$ initial value) |
| SPSA | Bisection steps=6, Upper-bound for bisection = 50.0, spsa-iters=1, spsa-samples=128, Lower-bound=0.001 Iterations = 100, lr=0.01,dr=0.01 |
| BIM (MNIST) | eps-iter=0.010 |
| BIM (CIFAR) | eps-iter=0.005 |

Table 12: Attack hyperparameters corresponding to Tables 4,6.

## APPENDIX H   ADAPTIVE-PGD

To further investigate *NeuralFP*'s dependence on gradient obfuscation, we evaluate the detection performance on $l_0$ and $l_2$ variants of an adaptive attack with projected gradient descent (PGD) (Madry et al., 2017) as the optimization method. The adaptive attack is set-up in a similar manner to adaptive-FGSM. PGD attack fails across all sets of hyperparameters explored (See Table 13). Even at distortion bounds of $\|\eta\|_\infty \leq 1.0$ and $\|\eta\|_2 \leq 60.0$ PGD does not succeed in degrading the performance of *NeuralFP*. Obviously, at these large distortion bounds (unbounded), adversarial examples exist as part of the training dataset itself. This indicates that the optimization procedure is not performing optimally, and it remains an open question if optimization procedures can be developed to render *NeuralFP* vulnerable.

| Method | Variant | Distortion Bound | Restarts | Iters | eps-iter | bisection steps | AUC-ROC (%) |
|---|---|---|---|---|---|---|---|
| PGD | $l_0$ | 16/255 | 1 | 1000 | 0.005 | 6 | 99.74 |
| PGD | $l_0$ | 16/255 | 50 | 50 | 0.005 | 6 | 99.21 |
| PGD | $l_0$ | 0.25 | 1 | 1000 | 0.005 | 6 | 99.71 |
| PGD | $l_0$ | 1.0 | 5 | 150 | 0.005 | 6 | 99.48 |
| PGD | $l_2$ | 10.0 | 5 | 100 | 0.5 | 6 | 99.55 |
| PGD | $l_2$ | 60.0 | 5 | 150 | 0.5 | 6 | 99.37 |

Table 13: The detection AUC-ROC for *NeuralFP* against *adaptive-PGD* (112 *pre-test* samples each). *NeuralFP* achieves near perfect AUC-ROC scores across settings. For CIFAR-10, the hyperparameters for *NeuralFP* are $N = 30, \varepsilon = 0.003$.

## APPENDIX I   ADAPTIVE-CW-$L_2$ - MORE ITERATIONS, MORE BISECTION STEPS

As a sanity check to see if the defense only succeeds because of the limited number of iteration steps during our preliminary investigation, we run Adaptive-CW-$L_2$ for 20000 iteration steps with 20 bisection search steps for the value of $\gamma_1$ and 10 bisection search steps for the value of $\gamma_2$. For a batch of 16 samples, this takes approximately 10 hours on an NVIDIA Tesla K80 for CIFAR-10 with $N = 30$. We find that even with these large number of iterations/bisection steps, *NeuralFP* is quite robust and the AUC-ROC does not degrade substantially. We evaluate Adaptive-CW-$L_2$ with these hyperparameters for two sets of *NeuralFP* hyperparameters. Table 14 summarizes the results from these experiments. Note that since the number of samples for these evaluations is quite small, the

AUC-ROC often fluctuates a few percent depending on the set of samples randomly chosen from the unseen test-set. Increasing the number of iterations causes a drop in AUC-ROC of about 2-4% when moving from 1000 iteration steps to 20000 steps. For *NeuralFP* with ($N = 50, \varepsilon = 0.003$), the AUC-ROC corresponding to the settings in Table 12 is 96.2%. An interesting observation is that the perturbation size (mean $l_2$ distortion) seems to grow with the number of increased iteration steps. For the vanilla CW-$L_2$ attack (non-adaptive), the mean $l_2$ distortion is in the range $0.15 - 0.25$. For further comparison, the defense in (Madry et al., 2017) has a robust-prediction accuracy of $< 5\%$ at a $l_2$ distortion of $0.45$ when evaluated against PGD.

| *NeuralFP* Parameters | Iteration Steps | Bisection Steps ($\gamma_1$) | Bisection Steps ($\gamma_2$) | Mean $l_2$ Distortion Size | Number of samples | AUC-ROC |
|---|---|---|---|---|---|---|
| $N = 30, \varepsilon = 0.003$ | 20000 | 20 | 10 | 0.47 | 112 | 94.12 |
| $N = 30, \varepsilon = 0.003$ | 15000 | 15 | 9 | 0.31 | 64 | 95.48 |
| $N = 50, \varepsilon = 0.003$ | 20000 | 20 | 10 | 0.45 | 64 | 95.56 |

Table 14: The detection AUC-ROC for *NeuralFP* against Adaptive-CW-$L_2$ with varying hyperparameter settings.

## APPENDIX J  ADAPTIVE-SPSA WITH 1000 ITERATIONS

For SPSA we investigate if increasing the number of iterations degrades performance for 3 sets of hyperparameters (See Table 15) with randomly sampled 112 unseen test-samples. We find that even with increased iterations the detection performance largely remains unaffected.

| Iteration Steps | Bisection Steps | Distortion Bound | Learning Rate | Delta | AUC-ROC (%) |
|---|---|---|---|---|---|
| 1000 | 20 | 0.05 | 0.01 | 0.01 | 99.84 |
| 1000 | 20 | 0.05 | 0.005 | 0.005 | 99.71 |
| 1000 | 20 | 0.25 | 0.01 | 0.01 | 99.32 |

Table 15: The detection AUC-ROC for *NeuralFP* against SPSA with varying hyperparameter settings.

## APPENDIX K  RANDOMIZED SAMPLING TO CHECK FOR GRADIENT OBFUSCATION

A test suggested in (Athalye et al., 2018) to check for gradient-masking is sampling randomly in $\epsilon$-balls around unseen test-data to find adversarial examples. They recommend sampling $10^5$ points or more, and checking if any of these points can pass through the defense. Accordingly, for 100 MNIST test samples, we sample $10^6$ points uniformly from the $l_\infty$-ball with $\|\eta\|_\infty \leq 0.3$ and for 50 CIFAR-10 test samples, we sample $5 \times 10^5$ points uniformly from the $l_\infty$-ball with $\|\eta\|_\infty \leq 0.25$. Amongst these randomly sampled points for each test point, we compute the smallest $\min_i \left( L_{fp} \left( x, i, \xi; \theta \right) \right)$ (let us call this value $L_{fp}^*$) across points that are adversarial (i.e. cause a misclassification). Note that $L_{fp}^*$ for each test point is the minimum fingerprint-loss over all adversarial samples sampled around the point and across all labels. For CIFAR-10, we are able to find adversarial examples for 47 points and for MNIST, we are able to find adversarial examples for 52 points. Figure 10 shows the distribution of $L_{fp}$ across the test points and $L_{fp}^*$ across the corresponding randomly sampled adversarial points. We observe that for MNIST, one single test sample has a high fingerprint-loss while all adversarial samples have high fingerprint-losses. The remaining test-samples have fingerprint-losses that are roughly $3 - 30$ times smaller than the adversarial examples. For CIFAR-10, the test and adversarial points are well separated, with most test samples having losses significantly smaller (roughly 10 times smaller) than the randomly sampled adversarial examples. This indicates that it is likely that *NeuralFP* does not simply function by misleading the gradient based attacks.

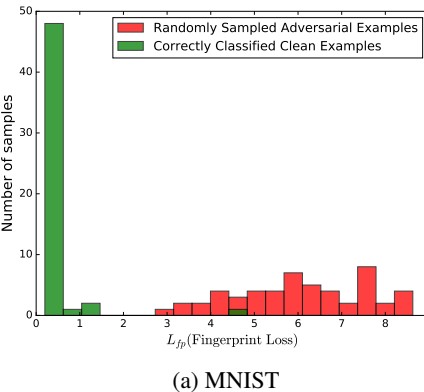

(a) MNIST

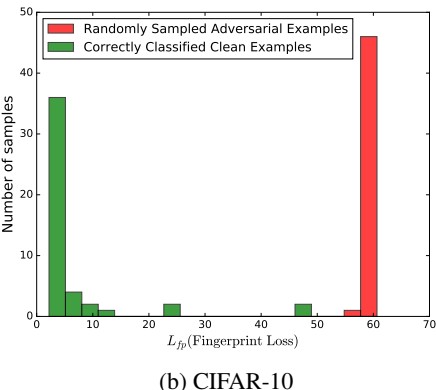

(b) CIFAR-10

Figure 10: Histograms depicting the distribution of losses for randomly sampled adversarial examples and test-data. Randomly sampled adversarial examples are well separated from unseen test examples. For *NeuralFP*, hyperparameters are $(\varepsilon, N) = (0.1, 10)$ and $(\varepsilon, N) = (0.003, 30)$ for MNIST and CIFAR-10 respectively.

