# OpenReview forum: "Detecting Adversarial Examples Via Neural Fingerprinting"
_ICLR.cc/2019/Conference_

### Official Review · AnonReviewer2 · 2018-11-04
**On the adaptive CW attack**

**Rating:** 6
**Confidence:** 3

**Review:**

This work introduces a novel defense method "Neural Fingerprinting" against adversarial examples.
In the training process, this method embeds a set of characteristic labeled samples so that responses of the model around real data show a specific pattern.  The defender can detect if a given query is adversarial or not by checking the pattern at test time.

Strong point:
The strong point is that the proposed method seems to be appropriate and technically original. The performance is well investigated and compared with several competitors.  The organization is good and the idea is clearly stated.

Weak point:
One question is that why the proposed method can be protective against the adaptive CW attack. In the public discussion, the authors mention that the defense works successfully because the landscape of the fingerprinting loss is non-convex and no gradient method is guaranteed to find a suitable solution. If this is correct, did you repeatedly try the gradient-based attack with changing random seeds? By doing so, the attack might work successfully with a certain probability.

Comments:
The presented method seems to have a certain similarity with digital watermarking of deep neural networks, for example:
https://gzs715.github.io/pubs/WATERMARK_ASIACCS18.pdf
It would be interesting to mention to these methods in the related work section.

---

> ### Author Response · Authors · 2018-11-11
> **Response to Reviewer 2**
>
> Thank you for your feedback. As detailed in our response to Reviewer 1, we are working on new experiments with randomized restarts for PGD with *many* iterations. We will report back with the numbers for our experiments with randomized restarts soon. Please see our response to Reviewer 1 for a detailed discussion with regard to the details of our new experiments and specific comments on gradient masking.
>
> Even though our loss is non-convex, the fingerprint loss L_{fp} is differentiable and continuous, and it is not obvious to us that any of the attacks we study are likely to succeed against NeuralFP, even with a larger distortion budget. We believe this is because finding “realistic” images in a high-dimensional space starting from the vicinity of the current image may not be an easy task .
>
> Besides the experiments with randomized restarts, which we are currently running, if you have further specific suggestions to improve the paper and the evaluation, we would be glad to hear them.
>
> We would like to also point out that the Carlini-Wagner attack (https://arxiv.org/abs/1608.04644) was compared with that of exhaustive search using a solver-based approach (https://arxiv.org/abs/1709.10207). It was shown that the Carlini-Wagner attack does not necessarily succeed in finding the optimal attack. Note that exhaustive search comes with tremendous computation costs (the problem is NP-hard!). This implies that there is no guarantee that the CW attack does indeed find the global minima, and finding the global optima through exhaustive search might be intractable.
>
> Also, we would like to note again that the SPSA attack and black-box attacks have been shown to overcome defenses that mask gradients. However, NeuralFP performs quite well against these attacks.

---

> ### Author Response · Authors · 2018-12-03
> **Revision with random seeds/many more iterations**
>
> Dear Reviewer,
> We would appreciate additional feedback on the new experiments (with more iterations/random restarts) as requested. We hope the following points and our revised paper address your concerns stated above.
>
> 1. We have run adaptive-PGD (which is a strong gradient based attack) with random restarts and many iterations (Appendix H), and adaptive-CW-l2 with *many* iterations (20000) (Appendix I), and see that NeuralFP is robust even for these settings.
>
> 2. CW-l2 attack has been shown to fail, even when there exist adversarial examples for certain. For example, in the paper https://arxiv.org/abs/1802.00420 (see Appendix) show that against Defense-GAN the attack is only able to reduce the accuracy to 55% even though adversarial examples exist on the projection manifold. As such, there is no formal guarantee that the CW-l2 attack will succeed.
>
> This is also discussed in their ICML talk ( https://nicholas.carlini.com/talks/2018_icml_obfuscatedgradients.mp4 ) at minute 15:00.

---

### Official Review · AnonReviewer3 · 2018-11-05
**Innovative perturbation-based learning strategy leads to very impressive performance in adversarial detection**

**Rating:** 9
**Confidence:** 4

**Review:**

This paper proposes a method for the detection of adversarial examples via what the authors term "neural fingerprinting" (NeuralFP). Essentially, a reference collection of perturbations are applied to the training data so as to learn the effects on the classification decision. The premise here is that on average, normal examples from a common class would have similar changes in the classification decision when reference perturbations are applied, whereas adversarial examples (particularly those off the local submanifold) may have a markedly different set of changes from what was expected for the targeted class. These reference perturbations as well as the anticipated output perturbations together form the "fingerprints".

To measure the difference between observed outputs and fingerprints, the average (squared?) Euclidean distance is used. Given a fixed set of input fingerprints (presumably chosen so as to provide coverage of the range of possible perturbation directions), the authors use the distance formula as a regression loss ("fingerprint loss") to train the choice of output fingerprints. Although the authors do not explicitly state it this way, this secondary training objective encourages a K-Means style clustering of output perturbations where the output fingerprints serve as cluster representatives.

This learning formulation is to my mind is both very innovative and extremely effective, as demonstrated by the authors' experimental results. Their experiments show superlative performance (near perfect detection!) against essentially the full range of state-of-the-art attacks. They give careful attention to the mode of attack, and show excellent performance even for adaptive white-box attacks, in which existing attack methods are given the opportunity to minimize the fingerprint loss.

The presentation of the paper is excellent - clear, well-motivated, and detailed, with careful attention given to experimental concerns such as the choice of perturbation directions (the recommendation is to choose them at random), and the number of fingerprints to pick.

Overall, the reported results are so good, and the approach so convincing, that one wonders what the weaknesses of the approach might be (if any). Questions that do come to mind are:
* Can an adversarial strategy can be developed that could execute a successful attack while minimizing the fingerprint loss.
* Another issue is whether the NeuralFP would work on more challenging data sets where the classes are highly fragmented - at what rate would the benefits of NeuralFP fade as the classification performance degrades?
* What happens to performance if the perturbation directions are chosen so as to better conform with the local sub-manifolds... would fewer perturbations be required? (It would seem that reducing the number of perturbations needed could have a significant effect on training time.)

Overall, this is a very strong and important result, fully deserving of acceptance.

P.S. Two sets of typos that need attention:
* In Equation 3, the Euclidean norm is taken. In Equation 5, the squared Euclidean norm is taken. Presumably, one of these is a typo. Which?
* In the definition of delta-min and delta-max in the first paragraph of Section 2.2, y-hat should be w-hat.

---

> ### Author Response · Authors · 2018-11-11
> **Response to Reviewer 3**
>
> Thank you for the comments and the feedback, we are glad you found our contributions interesting and promising.
>
> Addressing your questions:
>
> -- “Can an adversarial strategy be developed that could execute a successful attack while minimizing the fingerprint loss.”
>
> This is an interesting question, if one can indeed develop a new attack to break NeuralFP. We are actively investigating this, and we believe that NeuralFP will result in better attacks being formulated and eventually, better defenses.
>
> -- “Another issue is whether the NeuralFP would work on more challenging data sets where the classes are highly fragmented - at what rate would the benefits of NeuralFP fade as the classification performance degrades?”
>
> From the few experiments we report on MiniImagenet-20, we see good detection rates on FGSM and BIM. Also, when moving from MNIST to CIFAR-10, the classification performance degrades quite a bit, while NeuralFP’s performance nearly stays the same. We think these are promising signs that NeuralFP would be beneficial even on fragmented data-sets where classification is harder. However, conclusively answering this question concretely would need a thorough investigation.
>
> -- “What happens to performance if the perturbation directions are chosen so as to better conform with the local sub-manifolds... would fewer perturbations be required? (It would seem that reducing the number of perturbations needed could have a significant effect on training time.)”
>
> The increased training time (and memory) is indeed one of the biggest drawbacks of our approach, and the suggested idea of using a more principled approach for choosing the perturbation directions could indeed result in faster training, and possibly, better detection of outliers. We believe this is the immediate next step towards making NeuralFP more accessible on more complex tasks like ImageNet.
>
> Thanks for pointing out the connection to K-means clustering, that is an insightful interpretation. We will consider this and see if it sheds more light on our approach, and helps us improve NeuralFP.

---

### Official Review · AnonReviewer1 · 2018-11-06

**Rating:** 5
**Confidence:** 4

**Review:**

This paper proposes a new technique for detecting adversarial examples by introducing "fingerprints" into the landscape while training, and exploiting the fingerprints at test time to detect adversarial examples. The idea is novel and the paper is well-written, but concerns about gradient masking prevent me from recommending acceptance just yet.

Positives:
The paper is extremely well-written, and the approach is clear and presented well. The authors also clearly put significant effort into the evaluation, and accurately/consistently describe threat models that they consider. The approach is also clearly novel, and is interesting.

Concerns:
My biggest concern is that this detection mechanism masks gradients in its loss function. The two reasons I strongly believe this is the case are (a) Figure 5 and (b) the authors themselves state that their loss is highly non-convex and that no gradient-based method may be able to find a solution. This, however, does not guarantee robustness (see [1] for why such “unfriendly” landscapes can usually be circumvented)

Some concrete evaluation concerns and experiments that the authors can run to alleviate them:
- Figure 5 shows adversarial robustness even against eta = 0.25---at this value of epsilon, the attacker should be able to create realistic images of other classes (even without PGD), so this suggests that the loss is somehow making examples hard to find rather than removing them. The authors should address this issue.
- Showing that at a sufficiently high eta attacks start to succeed is also useful
- Running SPSA, CarliniL2-FP, and PGD for *many* more iterations and using *many* more steps for binary search (right now it looks like the binary search is looking in a space of size 10^6 with 10 steps, which only has a granularity of about 5k, which means you never see any value < 5k in a bisection search, which casts into doubt all of these results)
- The AUC should monotonically degrade with eta (this is another indication the attacks might not be running for long enough)
- The method does not seem to be specific to L-infinity constraint. To this end, a version of Figure 5 in the L2 case would be extremely useful in understanding the detection method.

I also apologize if some of these concerns about gradient masking seem unsubstantiated; that said, I tried to run the code given in the paper, but got several OOM and other errors (utils modules not found, and PyTorch deprecations), even on a machine with 8 12GB-memory GPUs. If the authors can provide instructions for running the code I will be happy to test it and alleviate some of my own concerns. I also tried to reimplement the approach, but did not manage to finish before the review deadline. If I am able to reimplement the approach I will update my review accordingly.

Some smaller comments on the paper:
A consolidated set of tables for attack parameters in an appendix is needed
- Page 4 last paragraph line 4 find the subset that “satisfies” instead of “satisfy”
- Page 5 paragraph 1 line 1 defender,for needs a space before for
- Page 5 paragraph right before theorem 1 last line Here, for detection needs a , after detection
- Page 7 paragraph 1 line 2 (2 hidden layers the 2 should be written two
- Page 7 second last paragraph line 2 “is chosen” instead of “are chosen”
- Page 7 last paragraph line 2 “across attacks” needs a , after
- Page 9 table 6 label line 2 “does not shown” should be “show” instead of “shown”
- Page 9 last line “measure of robustness” remove "of"

[1] https://arxiv.org/pdf/1802.00420.pdf

---

> ### Author Response · Authors · 2018-11-06
> **Help Running Our Code**
>
> Before, we address your other issues we would like to immediately get you to be able to run our code. We run our code on a machine with 2 GPUs, so the OOM error is quite surprising.
>
> Could you please let us know if you have access to AWS instances? We have been able to run our experiments on the AWS DeepLearning AMI. I can make a public AMI and share it with you if that's easier.
>
>
> I am able to run our code for MNIST on an Ubuntu 16.04 machine with a 16GB RAM (no GPU). The commands I am running on the Ubuntu machine are as follows:
> 1. cd into the directory with run.sh after extracting the files from the tar-ball
> 2. Run the command 'export PYTHONPATH=$(pwd):$PYTHONPATH'
> 3.  Run the command './run.sh mnist train attack eval nogrid 50 0.1 10'
>
> Also, this runs with python 2.7 and we have not tested with python 3 or above.
>
> We have tested our code on multiple linux platforms with the following specific software versions:
>
> >>> torch.__version__
> '0.3.0.post4'
>
> >>> torchvision.__version__
> '0.2.0'
>
> >>> tensorflow
> '1.4.0'
>
> >>> keras
> '2.1.0'
>
> We are actively currently working on running PGD (with random restarts) and SPSA with many more iterations. We will provide detailed answers to your questions, and do our best to get you running with the code ASAP! Engineering the whole project is considerable effort, we strongly recommend trying to run our scripts which we have shared.

---

> > ### Comment · AnonReviewer1 · 2018-11-06
> > **Thank you**
> >
> > Thanks! I have started a thread which is only visible by the authors, in order to keep the page limited to discussion about the content of the paper rather than technical help.

---

> > > ### Public Comment · ~Nicholas_Carlini1 · 2018-11-07
> > > **Please don't count this against the paper**
> > >
> > > A quick unsolicited comment for reviewer 1: please don't penalize the authors for any difficulties you have in running their code. This is one of two adversarial example defense paper at ICLR this year that has released code with their paper (I believe), and I would hate to see them penalized for it.
> > >
> > > I fear one of the main reasons authors don't release code is that it's currently seen only as a potential liability (what if someone breaks my model faster because I released the code? what if someone tries to run it and it doesn't work? etc) rather than a strength (now people can reproduce my work/verify my claims/etc).
> > >
> > > Eventually I'd hope that providing code with papers becomes so common that criticizing a paper because the code is hard to work with will be a valid critique. But I don't think we're there yet. The fact that the authors released the code at all is commendable.

---

> > > > ### Comment · AnonReviewer1 · 2018-11-07
> > > > **I am not counting this against the paper in the slightest**
> > > >
> > > > As your comment mentions, it is commendable that the authors released their code at all, and I actually counted this in favor of the paper in my review. I only said that I couldn't run their code in order to justify the fact that my concerns about gradient masking were not substantiated via my own testing (ideally, rather than suggesting that the defense masks gradients, I could have run some evaluations myself and assessed this). I am now privately communicating with the authors in order to get the code up and running.
> > > >
> > > > Again, for me the fact that the code is released (and in general, the authors' commitment to openness and reproducibility) is impressive and definitely contributed positively towards my impression of the paper.

---

> > > > ### Author Response · Authors · 2018-11-07
> > > > **Thanks! We are committed to openness about our defense!**
> > > >
> > > > We have shared one set of well performing model weights, the defense parameters in the link below (and also, some adversarial examples we generated as well).
> > > >
> > > > https://www.dropbox.com/sh/0ow62124skbzy1y/AABOHL0Y0EeAdEmPfHItpOt_a?dl=0
> > > >
> > > >
> > > > Thank you for the comment!  We are actively working with the reviewer to get them to be able to run our code, and will support anyone else who wishes to do the same during the review process.

---

> ### Author Response · Authors · 2018-11-11
> **Response to review (part 1)**
>
> We sincerely thank the reviewer for taking time to implement the paper and run our experimental code,  we believe that openness of any defense is a key necessary step in the evaluation process and we will do our best to support your efforts.
>
> While we (and possibly, the reviewer) run experiments with a large number of iterations, with large number of steps for the bisection search (and for PGD, *many* randomized restarts), to encourage discussion we are posting an initial response addressing the reviewer’s concerns.
>
> *** Regarding gradient masking:
>
> First, we would like to address your main concern, i.e., gradient masking.
>
> Other than the fact that FGSM and SPSA fail with large eta, NeuralFP does not show any other signs of gradient masking. The reference [1] mentions a number of features of gradient masking. We note that NeuralFP behaves *contrary* to defenses that rely on gradient masking in many ways: it is
>
> 1) robust to black-box attacks, SPSA attack and adaptive attacks, and
> 2) single step attacks do worse than multi-step attacks.
>
> This indicates that NeuralFP may not rely on gradient-masking. With the current set of available attacks it seems hard to conclude (beyond reasonable doubt) about NeuralFP’s reliance on gradient masking, one way or another.
>
> “ The two reasons I strongly believe this is the case are (a) Figure 5 and (b) the authors themselves state that their loss is highly non-convex and that no gradient-based method may be able to find a solution. This, however, does not guarantee robustness (see [1] for why such “unfriendly” landscapes can usually be circumvented)”
>
> First we note that the function L_{fp} is continuous and differentiable with respect to the logits. Also, please note all of the attacks used in [1] are indeed gradient based.
>
> Second, [1] (See section 4) introduces three strategies to traverse “unfriendly” landscapes:
>   1. BPDA
>   2. Expectation over transformation
>   3. Reparameterization
>
> However, we note that none of these techniques seem directly applicable to our defense:
> a) BPDA is applicable in settings where gradient information is masked with either non-differentiable operations or optimization procedures that are likely to cause the gradients to not be readily available. Our loss is continuous and differentiable. This leads us to believe BPDA is not relevant to this setting.
> b) Expectation over transformation -- This mechanism is not relevant as NeuralFP involves no randomization during detection.
> c) Reparameterization -- There is no reparameterization involved in the defense. The network *directly* acts on the image space.
>
> Phenomena such as label leaking (https://openreview.net/forum?id=BJm4T4Kgx) are not likely because we do not train using any information about adversarial examples.
>
> However, we understand that it might be possible that there exist attacks that break our defense since we do not provide formal guarantees. We have conducted experiments to extensively evaluate our defense, and are continuing to work on evaluating the defense in more detail.
>
> In this light, NeuralFP sheds light on possible weak spots in several of the attacks we evaluated against. This implies that either NeuralFP is indeed strong and/or we need stronger attacks for performing accurate evaluations -- both situations being beneficial to the adversarial ML community.  We are happy to summarize these discussion points in the final paper.

---

> > ### Author Response · Authors · 2018-11-11
> > **Response to Review (part 2)**
> >
> > *** Extra experiments:
> >
> > To alleviate your concerns regarding attacks with more iterations, we are running more experiments as suggested. Note that our previous experiments were already extensive -- as noted by the reviewer, we have put in significant effort into our evaluation.
> >
> > For 112 unseen CIFAR-10 test points we are running:
> > 1. CW-fp with 20,000 iterations and bisection search with 25 steps for gamma-1 and 12 steps for gamma-2 (roughly 6,000,000 gradient steps per point!).
> >
> > 2. Adaptive-PGD attack (similar to the other adaptive attacks) with
> > a) 1,000 steps and 1 randomized restart with 6 bisection steps, eps = 16.0/255.0, eps_iter=0.005 (same parameters as https://arxiv.org/pdf/1807.10272.pdf)
> > b) 50 steps and 50 randomized restarts with 6 bisection steps, eps = 16.0/255.0, eps_iter=0.005
> > We believe 6 bisection steps suffice here because we are searching for gamma in the range (0.01,10) -- at gamma>10, misclassification starts to fail for a significant fraction of the samples. The fact that increasing gamma causes misclassification to fail indicates that the gradients from L_{fp} are likely not exploding or vanishing.
> >
> > 3. We ran SPSA with 1,000 iterations and 20 bisection steps (with default parameters for eps=0.05, lr=0.01 and delta=0.01). The AUC-ROC is 99.97%. We are running experiments with
> > 1. lr=0.005 and delta=0.005 to see if that leads to better performance.
> > 2. eps=0.25, 1000 iterations, lr=0.01 and delta=0.01 to see if SPSA degrades accuracy at larger eta.
> >
> > Once we have all the numbers, we will compile the results and update our paper and the page. If you would like to suggest other relevant experiments, please let us know.
> >
> > *** Specific comments:
> >
> > -- “Figure 5 shows adversarial robustness even against eta = 0.25….. hard to find rather than removing them. The authors should address this issue.”
> >
> > We have not seen any paper that has been able to generate “realistic” images (for e.g. starting with a ‘2’ and ending up with a ‘6’) using any of the attacks we test against. It would be very useful if you could point us to related work that has been successful in doing so. We believe traversing the manifold of natural images in a high-dimensional spaces is an inherently challenging optimization problem (assuming a reasonable compute budget). There is no guarantee that any of the current attack techniques we are aware of will find other realistic images in a computationally tractable way (e.g., within a reasonable number of iterations/random restarts), starting from the vicinity of one image.
> >
> > -- “-Showing that at a sufficiently high eta attacks start to succeed is also useful”
> >
> > This makes sense in the context of robust prediction, where for large perturbations you can turn realistic images into noise where classification makes no sense. The functioning of our method is entirely different to robust prediction methods, since noise is unlikely to pass through NeuralFP. One way to definitely fool NeuralFP is to generate images from the true data-distribution, but as discussed, generating “realistic” images from the true data-distribution is not necessarily an easy task.

---

> > > ### Author Response · Authors · 2018-11-11
> > > **Response to review (part 3)**
> > >
> > > -- “- Running SPSA, CarliniL2-FP, and PGD for *many* more iterations and using *many* …… these results)”
> > >
> > > *** Regarding Bisection Search:
> > > We use a starting point of 1e-2 (<5k) for gamma (please see our hyper-parameters at the end of the public discussion) during the bisection search. Please note that this same bisection search is used in the original CW-L2 attack. Further, larger the gamma, more preference is given to minimizing L_{fp}.
> > >
> > >
> > > *** Regarding SPSA:
> > >
> > > We directly inquired with the authors of (https://arxiv.org/abs/1802.05666) about choosing the optimal Lagrange parameter to use for the SPSA attack. We were informed that the Lagrange coefficients for their experiments were hand-picked loosely based on the values of the losses. They further recommended that the simplest thing to do, in the context of their paper, is to just optimize J = C * min(1, J_adv) + L_CNN (for large C). This will cause the optimizer to ensure the image is misclassified, but stop optimizing the misclassification loss once it is already misclassified, and then focus on the likelihood loss. In summary, we believe the Lagrange parameter itself does not affect the results to that great an extent, and a coarse search should suffice. However, we are doing a finer search now as detailed at the end of our comment.
> > >
> > > *** Regarding CarliniL2-FP:
> > >
> > > Note that the loss in CW-fp has a similar form as that used in the SPSA paper (the loss in the SPSA paper was modelled after the loss from the CW attack), hence we believe that the optimization of L_{CW} and L_fp is not too sensitive to the exact Lagrange coefficients; a coarse search should suffice in this case as well. In the paper where the CW-loss function was introduced, the authors argue that the coefficient matters the most in producing the smaller adversarial perturbation. However, as suggested by the reviewer, we are running experiments with an increased number of bisection steps and iterations to be sure of the result.
> > >
> > > *** Regarding PGD:
> > > We are currently running PGD with a large number of iterations and restarts as discussed above. We will post the results shortly.
> > >
> > > -- “The AUC should monotonically degrade with eta (this is another indication the attacks might not be running for long enough)”
> > >
> > > We agree that with large eta this statement holds true for robust prediction, but it is not obvious that this should indeed be the case for *detection* with NeuralFP, as explained earlier. Large eta can be used to generate noisy images, but it is not entirely obvious that one would be able to generate realistic images from the true distribution by simply allowing for a large eta using the current attack schemes in a computationally tractable manner.
> > >
> > > --  “The method does not seem to be specific to L-infinity constraint. To this end, a version of Figure 5 in the L2 case would be extremely useful in understanding the detection method.”
> > >
> > > Indeed, our method is not specific to any adversarial attack. We test against the JSMA attack, which is L-0 based and we have near perfect detection. The CW attack attempts to minimize the L-2 perturbation and we cannot vary the L-2 perturbation as a hyperparameter. Do you recommend any specific L2 based attacks to test-against?
> > >
> > > [1] https://arxiv.org/pdf/1802.00420.pdf

---

> > > > ### Comment · AnonReviewer1 · 2018-11-11
> > > > **Response (part 1)**
> > > >
> > > > Thank you for the in-depth reply to the review. Although it will take me some time to fully read and respond to the rebuttal, there are some crucial points that I think may inform the authors' revisions:
> > > >
> > > > - R.e. papers that turn 2s into 6s etc, see this paper: https://arxiv.org/abs/1805.12152, which I believe may be useful.
> > > >
> > > > - R.e. high eta leading to failures: this should actually be true both for detection and prediction. For example, at eta = 1.0, an attack that selects the nearest neighbor from another class in the training set should suffice to get 0% detection. In the same vein, an attack that simply uses a generative model of some kind to find images of a class while minimizing distance to an original image should suffice to find high-eta examples of other images that are not adversarially perturbed in the first place.
> > > >
> > > > In any case, if extending Figure 5 to eta = 1.0 still yields a high success rate at detection with the current attacks, this indicates that it is a failure of current optimization methods, rather than a true detection mechanism, that might be generating this success, so this would be useful. It is then unclear if this inability is due to computational hardness, or simply an extremely non-convex landscape that requires "tricks" to be able to navigate.
> > > >
> > > > - PGD using L2-normalized gradients is an L2 attack which does not minimize l2 distance (i.e. taking gradient steps (step size) * grad / ||grad||. (In the same vein, the authors should also verify that the L-inf PGD attacks they are using normalize the gradient via sign, i.e. (step size) * sign(grad).)
> > > >
> > > > Concretely, it would be very helpful for the authors to show:
> > > > - Figure 5 for Linf and L2 (using PGD [this means sign(grad) and grad/||grad|| respectively] in both cases), extending from eta=0 to eta=1.
> > > >
> > > > I will read and address the rest of the points directly, but just wanted to respond to these in order to give reasonable time for experimentation. Thanks again for the detailed response, and the overall thoroughness.

---

> > > > > ### Author Response · Authors · 2018-11-16
> > > > > **Response to response (part 1)**
> > > > >
> > > > > Thanks for the quick reply and the heads up about the experiments.
> > > > >
> > > > > “- R.e. papers that turn 2s into 6s etc, see this paper:     https://arxiv.org/abs/1805.12152, which I believe may be useful.”
> > > > >
> > > > >
> > > > > Thanks for this interesting reference. We respectfully note that the adversarial  examples illustrated in the paper can clearly be deciphered to be altered and unnatural images even with the human eye (e.g. Figure 3 in the paper). This by itself is not convincing evidence that PGD/SPSA can generate “realistic” images from the “true” distribution.
> > > > >
> > > > > “- R.e. high eta leading to failures: this should actually be true both for detection and prediction. For example, at eta = 1.0, an attack that selects the nearest neighbor from another class in the training set should suffice to get 0% detection. In the same vein, an attack that simply uses a generative model of some kind to find images of a class while minimizing distance to an original image should suffice to find high-eta examples of other images that are not adversarially perturbed in the first place.”
> > > > >
> > > > > We agree that “realistic” unbounded attacks undoubtedly exist, e.g., images from the training dataset. We would like to clarify that most of our earlier comments with regards to finding realistic images in the rebuttal were in regard to the attacks that we use to  evaluate NeuralFP, such as SPSA/PGD/FGSM/CW.
> > > > >
> > > > > We would like to emphasize that our contribution focuses on detecting *bounded* attacks from the literature. Note that we do *not* evaluate or claim robust detection of (yet unpublished) unbounded attacks, e.g., based on generative models such as GANs (e.g., Song et al, NIPS 2018  https://arxiv.org/abs/1805.07894). Although we plan to evaluate on detection of unbounded (generative) attacks shortly, the scope of the current paper is limited to bounded adversarial attacks.
> > > > >
> > > > > Furthermore, for bounded adversarial attacks (e.g., eta<0.25), robust prediction is certainly a more difficult task than detection. For instance, Schmidt et al, NIPS 2018 (https://arxiv.org/abs/1804.11285) argue that robust prediction requires significantly larger amounts of data.
> > > > >
> > > > > “In any case, if extending Figure 5 to eta = 1.0 still yields a high success rate at detection with the current attacks, this indicates that it is a failure of current optimization methods, rather than a true detection mechanism, that might be generating this success, so this would be useful."
> > > > >
> > > > >
> > > > > We agree that failure at eta=1.0 would indicate flaws in the evaluated attacks, e.g., their inability might be due to the computationally hardness or might need improved optimization algorithms. We will try to address your suggestion to evaluate up to eta=1.0. However, we note that is computationally demanding.
> > > > >
> > > > > Note that for each eta, the PGD attack evaluation (with the bisection search  and 1000 iterations) for 112 samples is expected to take 4-5 days on a single GPU. We will try to run as many cases as possible before the revision dates, but we have a limited compute budget and time constraints.
> > > > >
> > > > > Furthermore, although evaluating at large eta is interesting, note that we do *not* claim NeuralFP can detect all (unbounded) adversarial attacks. However, given its excellent detection performance on both bounded gradient and non-gradient based attacks, we do believe it is a very strong baseline. As such, NeuralFP highlights the need for better attacks.
> > > > >
> > > > > Also, as we discussed in our previous replies, we do not believe our experimental results show that NeuralFP provides robustness merely through gradient masking. We feel that failure of the attacks at eta=1.0 also would not provide conclusive evidence of gradient masking, but rather a failure of the current attacks. However we are running experiments with larger eta-s and will update the manuscript/the page shortly.

---

> > > > > > ### Author Response · Authors · 2018-11-16
> > > > > > **Response to response (part 2)**
> > > > > >
> > > > > > “It is then unclear if this inability is due to computational hardness, or simply an extremely non-convex landscape that requires "tricks" to be able to navigate.”
> > > > > >
> > > > > > We feel that this an extremely interesting question, and we have extensively shown that the current set of optimization methods used to effectively evaluate adversarial attacks do not convincingly answer this question. Further (theoretical) analysis of this question is an exciting direction for future research.
> > > > > >
> > > > > > "- PGD using L2-normalized gradients is an L2 attack which does not minimize l2 distance (i.e. taking gradient steps (step size) * grad / ||grad||. (In the same vein, the authors should also verify that the L-inf PGD attacks they are using normalize the gradient via sign, i.e. (step size) * sign(grad).)
> > > > > >
> > > > > > Concretely, it would be very helpful for the authors to show:
> > > > > > - Figure 5 for Linf and L2 (using PGD [this means sign(grad) and grad/||grad|| respectively] in both cases), extending from eta=0 to eta=1."
> > > > > >
> > > > > > *** Attack normalization.
> > > > > > We use the default Cleverhans implementation for the SPSA/PGD attacks and (https://github.com/carlini/nn_robust_attacks)  for the CW attack. We did verify that Cleverhans normalizes the gradient via sign for PGD.
> > > > > >
> > > > > > *** Experiments and compute budget.
> > > > > > Thanks for the suggestions for new experiments. Running PGD (both versions) with the above mentioned #iterations/bisection steps, at a coarseness of \delta eta = 0.05, requires about 40 GPUs for 5 days, which equates to ~$10000 of AWS credits. Since we work in an academic setting, this is beyond our compute budget.
> > > > > >
> > > > > > We will try PGD with the normalized L2 norm and do our best within the allowed time frame and our financial constraints. However, we do feel our current extensive experiments already stand on their own as strong positive validation of NeuralFP.
> > > > > >
> > > > > > *** Additional tests for gradient masking
> > > > > > We are also working on an additional test for gradient masking with a large set of random-points described in [1]. We will update the page and our manuscript with these results shortly.
> > > > > >
> > > > > > Please let us know if you need any additional technical help with your evaluation and if you have other questions.

---

> > > > > > > ### Comment · AnonReviewer1 · 2018-11-16
> > > > > > > **Thank you!**
> > > > > > >
> > > > > > > Thank you for the reply. I am looking forward to the results of the other experiments (PGD >1000 iters, other experiments from the initial rebuttal).
> > > > > > >
> > > > > > > I understand if running the Figure 5 experiment again as suggested above would be too costly, but would it be possible to simply try the attack only with eps=1.0 for l-infinity, eps=10.0 for l2, and unbounded PGD for l2?
> > > > > > >
> > > > > > > Thanks again for the response; please let me know when the additional results (the standard ones from the initial rebuttal) are complete.

---

> > > > > > > > ### Author Response · Authors · 2018-11-16
> > > > > > > > **Clarification -- step-size**
> > > > > > > >
> > > > > > > > Minor clarification: Do you have a preferred step-size for the bounded and unbounded l2 PGD attacks? I am struggling to find literature/code that recommends a good step-size.
> > > > > > > >
> > > > > > > > Thanks again for the help and feedback!

---

> > > > > > > > > ### Comment · AnonReviewer1 · 2018-11-16
> > > > > > > > > **Hyperparameters**
> > > > > > > > >
> > > > > > > > > Ideally these could be found via grid search. Under computational constraints, however, I would try something like epsilon=0.5 for the l2 attacks (with l2 normalization).
> > > > > > > > >
> > > > > > > > > Good luck and thanks for the responsiveness.

---

> > > > > > > > ### Author Response · Authors · 2018-11-27
> > > > > > > > **Updated experiments**
> > > > > > > >
> > > > > > > > We have posted a general comment summarizing the results from our new experiments. Apologies for the delay as these experiments took a while to run, particularly CW-fp (with large iterations) and PGD with multiple restarts. We have also updated the paper to include these results (please see Appendix for detailed discussion/more numbers).
> > > > > > > >
> > > > > > > > We are happy to continue providing technical support with the implementation/models, and hope our experiments have been able to alleviate some of your concerns. Please let us know if you have further questions/comments!
> > > > > > > >
> > > > > > > > Many thanks for the review, and feedback on new experiments.

---

### Public Comment · (anonymous) · 2018-10-30
**Evaluation questions**

In Figure 5, it appears that AUC is uncorrelated with the magnitude of the adversarial perturbation. If you increase the perturbation budget to something much larger does the AUC begin to decrease? (Also: what is the dataset for this figure? MNIST?)

The adaptive CW attack is unbounded, and should always eventually succeed. That is, for any fixed decision threshold that could be used for detection, the attack should always eventually produce an adversarial example that can fool the detector by introducing a very large perturbation. However, it looks like you claim robustness even at very small thresholds (e.g., a 80+% TPR @ 0%FPR). Do you know what is going on here?

---

> ### Author Response · Authors · 2018-10-30
> **Addressing evaluation questions**
>
> Thank you for the questions, and catching the missing detail in Figure 5.
>
> “In Figure 5, it appears that AUC is uncorrelated with the magnitude of the adversarial perturbation. If you increase the perturbation budget to something much larger does the AUC begin to decrease? (Also: what is the dataset for this figure? MNIST?)”
>
> The dataset for Figure 5 is CIFAR-10 — we will include this detail in the paper. 0.25 is already a significantly large perturbation, since our pixel values in the input images are scaled from [-0.5,0.5]. We will run experiments with larger perturbations and report the numbers for this shortly.
>
> “The adaptive CW attack is unbounded, and should always eventually succeed. That is, for any fixed decision threshold that could be used for detection, the attack should always eventually produce an adversarial example that can fool the detector by introducing a very large perturbation. However, it looks like you claim robustness even at very small thresholds (e.g., a 80+% TPR @ 0%FPR). Do you know what is going on here?”
>
> The adaptive CW attack is not always guaranteed to work for an arbitrary detection rule for an arbitrary point. For example, the adaptive CW attacks formulated in [1,2] do not succeed. However, we do note that the adaptive CW attack fails in [1] due to gradient obfuscation and [2] has not been tested extensively for gradient obfuscation.
>
> For the adaptive CW attacks we consider, we do not set up a fixed threshold, instead we attempt to misclassify the point and minimize the fingerprint loss corresponding to the *true class*. From all the successful attacks that produce a misclassification during the iterations, we choose the one with the smallest fingerprint loss.
>
> Also, we would like to clarify the intuition: At smaller thresholds it is actually harder for the attacker to fool the detector. This is because the fingerprint loss under smaller thresholds admits smaller regions while larger thresholds admit larger regions.
>
> The fingerprints provide a characterization of the data-distribution and reject regions away from the data-distribution corresponding to the training data . It is not obvious if
> 1) there are points that successfully produce a misclassification and have a small fingerprint loss (corresponding to the true label), and
> 2) if the CW attack will find it.
>
> For instance, consider Figure 3 — there exist no “adversarial regions” that produce a misclassification and have a small fingerprint loss corresponding to the true label. We conjecture that the networks from our experiments on the image dataset (MNIST, CIFAR-10…) show similar behavior — the “adversarial regions” producing both a misclassification and a small fingerprint loss do not exist/are either quite small and difficult to find.
>
> We have also considered a variant where the adaptive CW attacker can determine both the target-class for the label-misclassification and the fingerprint loss to be minimized — constrained to both being the same. This attack performed worse than the variant we report in the paper.
>
> Our source code (link provided in the paper) includes the implementation for the adaptive CW attack discussed in the paper. If you would like to suggest other variants that are interesting, we would be happy to try them out.
>
> [1] https://openreview.net/forum?id=B1gJ1L2aW (ICLR’18)
> [2] https://arxiv.org/abs/1807.03888 (NIPS’18)

---

> > ### Public Comment · (anonymous) · 2018-10-30
> > **Concerned about gradient masking**
> >
> > It's concerning to me that the attack still is failing even at very large epsilons of 64/255 on CIFAR-10. No prior defense has achieved robustness at even 16/255. It might be useful to verify that the attack does eventually succeed if epsilon is allowed to be 128/255. (Not because this is an actual valid attack -- but just to make sure the attack is functioning correctly.)
> >
> > > It is not obvious if
> > > 1) there are points that successfully produce a misclassification and have a small fingerprint loss (corresponding to the true label), and
> >
> > These points definitely exist: we know that there exists at least one image correctly classified as each class, so given an unlimited perturbation budget, there will always exist an "adversarial example" with unbounded distortion.
> >
> > > 2) if the CW attack will find it.
> >
> > It may not. But clearly an optimal attack should find it, and so if detection rate is not 0% then we know the attack is performing sub-optimally.

---

> > > ### Author Response · Authors · 2018-10-30
> > > **Addressing comments on gradient masking**
> > >
> > > >It's concerning to me that the attack still is failing even at very large epsilons of 64/255 on CIFAR-10. No prior defense has achieved
> > > >robustness at even 16/255. It might be useful to verify that the attack does >eventually succeed if epsilon is allowed to be 128/255.
> > > >(Not because this is an actual valid attack -- but just
> > > >to make sure the attack is functioning correctly.)
> > >
> > >  The optimization problem of finding adversarial examples that both misclassify and have a low-fingerprint loss is non-convex, and no gradient method is guaranteed to find a suitable solution (i.e. good local minimum). Though we do not provide *formal* guarantees, it is not obvious if a “computationally tractable functional” attack exists that with increasing epsilon will fool neural fingerprinting. We have evaluated with a variety of currently known attacks, and find that neural-fingerprinting is robust across these attacks.
> > >
> > > To evaluate if the defense is simply based on gradient masking, we evaluate both with the adaptive-SPSA attack and black-box adversarial examples, and observe that the detection AUCs do not degrade. Please note that SPSA is gradient-free.
> > >
> > > Further, we do notice a small degradation in AUCs for the adaptive BIM-b and CW-L2, which suggests that the adaptive attacks are not entirely dysfunctional -- which would have likely been the case if the defense was working solely based on masked gradients.
> > >
> > > Another sign against gradient masking is that one step attacks do strictly worse than iterative attacks, unlike [1] cited above.
> > >
> > > >> It is not obvious if
> > > >>1) there are points that successfully produce a misclassification and have a small fingerprint loss (corresponding to the true label), and
> > >
> > > >These points definitely exist: we know that there exists at least one image correctly classified as each class,
> > > >so given an unlimited perturbation budget, there will always exist an "adversarial example" with
> > > >unbounded distortion.
> > >
> > >
> > > Thank you for pointing this out, we agree that with unbounded perturbations, such examples exist. For instance, an unbounded “successful attack” could transform a natural image of a dog to a natural image of a cat. But, coming up with an attack that finds such "successful" unbounded transformations  is likely hard.
> > >
> > > An intuition for why finding such “successful” unbounded transformation is likely hard might be that real images are thought to occupy a very low-dimensional subspace in the high-dimensional input space, and so the space of “good” perturbations might be a very small fraction of the space of all allowed perturbations that a CW attacker searches over.
> > >
> > > >> 2) if the CW attack will find it.
> > >
> > > >It may not. But clearly an optimal attack should find it, and so if detection rate is not 0% then we know the
> > > >attack is performing sub-optimally.
> > >
> > > As stated earlier, an optimal attack may be intractable. For reference, in our experiments for the CW-attack, we used the following settings:
> > >
> > > L2_BINARY_SEARCH_STEPS_1 = 9 (corresponding to gamma-1)
> > > L2_BINARY_SEARCH_STEPS_2 = 5 (corresponding to gamma-2)
> > > L2_MAX_ITERATIONS = 1000
> > > L2_ABORT_EARLY = True
> > > L2_LEARNING_RATE = 1e-2
> > > L2_TARGETED = True
> > > L2_CONFIDENCE = 0
> > > L2_INITIAL_CONST = 1e-3    (gamma-1 initial value)
> > > L2_INITIAL_CONST_2 = 0.1   (gamma-2 initial value)
> > >
> > > Meanwhile, we will run experiments with larger L2_MAX_ITERATIONS to see if there is a significant degradation in AUC-ROC. We welcome suggestions from you (and other readers) for better settings or modifications to consider.

---

> > > > ### Author Response · Authors · 2018-10-30
> > > > **SPSA parameters**
> > > >
> > > > Some additional information for the OP and the interested reader:
> > > >
> > > > The SPSA attack is designed for when the gradients do not point in useful directions -- essentially, to check for gradient obfuscation. Note that our method uses no information about the adversarial examples or the attack mechanism during train, so things like label leaking cannot even occur. For SPSA, we use the cleverhans implementation and the hyper-parameters we test against are:
> > > > delta=0.01
> > > > learning rate=0.01
> > > > iterations=100
> > > >
> > > > (Same as the parameters from the SPSA [3] paper used to attack the CIFAR-10 dataset).
> > > >
> > > > [3] https://arxiv.org/abs/1802.05666 (ICML'18)

---

> > > > ### Comment · AnonReviewer3 · 2018-11-26
> > > > **Non-convex optimization is forced?**
> > > >
> > > > > The optimization problem of finding adversarial examples that both
> > > > > misclassify and have a low-fingerprint loss is non-convex, and no gradient
> > > > > method is guaranteed to find a suitable solution (i.e. good local minimum).
> > > > > Though we do not provide *formal* guarantees, it is not obvious if a
> > > > > “computationally tractable functional” attack exists that with increasing
> > > > > epsilon will fool neural fingerprinting. We have evaluated with a variety of
> > > > > currently known attacks, and find that neural-fingerprinting is robust
> > > > > across these attacks.
> > > >
> > > > I think that this may be the key observation here. By forcing an inherently non-convex landscape on the adversary, it's not out of the question that gradient methods trained on smooth(er) approximations would fail against NeuralFP. I am not convinced that gradient masking is the underlying cause for the good performance reported by the authors.
> > > >
> > > > Kudos to Reviewer 1 for insisting that some diligence should be required by authors in demonstrating that their detection methods cannot be easily circumvented due to a reliance on gradient masking. Even more than the tests requested by R1, I would have liked to see BPDA included in the study, so as to ensure that the gradient approximations used by BPDA are not sufficient to get it unstuck from bad local minima in this non-convex landscape. In all their responses, their dismissal of BPDA is the one that ultimately concerns me the most.
> > > >
> > > > However, this paper greatly advances the debate. The unprecedented excellent performance of their adversarial detection strategy deserves some recognition, and with its non-convex optimization function, some benefit of the doubt regarding gradient masking.

---

> > > > > ### Comment · AnonReviewer3 · 2018-12-04
> > > > > **Fingerprinting makes gradient masking harder to expose**
> > > > >
> > > > > First, I'm glad to see that the authors have managed to run their experiments in time. I am not surprised to see that they still achieve high accuracy even for unbounded attacks.
> > > > >
> > > > > In my previous reply I stated that the authors are (at least implicitly) forcing an inherently non-convex landscape on the adversary. What I mean by this is that they are making the optimization complexity arbitrarily high by introducing these N fingerprint directions. In order to consistently break the defense, an attacker would need to consistently succeed in learning perturbations from x to x' that produce misclassifications. Although x' is close to x, it is expected to have its "secondary" perturbation direction deltas (one for each of the N fingerprint directions) conform to those of the new class (of x') rather than the old class (of x). This is an extremely non-convex problem whose difficulty of this is parameterized by N.  If N is high enough, even an unbounded attack would eventually succumb to this effect, due to the increase in accumulation of contributions in the N terms of the fingerprint loss.
> > > > >
> > > > > (The authors don't state their argument as forcefully as they could, but they do explain the computational burden faced by any attacker that might seek to reverse engineer the fingerprint directions, in the first few paragraphs of Section 2.1.)
> > > > >
> > > > > This paper constitutes a new "arms race": on the one hand, the adversary has the weapon of "exposing" gradient masking through loss approximation and other search or smoothing tricks; on the other hand, the defender can now (by raising the number of hidden "fingerprint" directions) increase the burden on the adversary. The authors' experimental results do show that increasing N improve the effectiveness of their defense. What is surprising is that N does not need to be very high to defeat the attacks they consider. Exploration of this tradeoff will be a great area for future investigation.

---

> > > > > > ### Author Response · Authors · 2018-12-05
> > > > > > **More discussion, numbers and intuition based on your comment**
> > > > > >
> > > > > > We agree that we could be more forceful about emphasizing the difficulty of the non-convex problem the adversary has to solve, we will revise to enforce this point during revision. We would like to further add two more points of discussion to your comment:
> > > > > >
> > > > > > 1. Using more fingerprints leads to more robustness to adaptive attacks, and poses a much harder optimization problem as you mentioned. We did find some evidence that directly points to this during our tests.
> > > > > >
> > > > > > For MNIST with hyper-parameters N=6, \vareps=0.1, against non-adaptive attacks NeuralFP achieves >95% AUC-ROC detection. When evaluated with adaptive-CW-l2, this number drops to 70% or less (with some variance due to the random dx chosen).
> > > > > >
> > > > > > With N=20, eps=0.05, for the non-adaptive attacks AUC-ROC is near 100% and when evaluated with the adaptive-CW-l2 attack AUC-ROC is still >95%. We see a similar effect with CIFAR-10 as well (in the appendix).
> > > > > >
> > > > > > 2. To provide extra intuition we discuss some observations from our experiments.
> > > > > >
> > > > > > Hypothesis: NFP has a flat loss-surface away from the data.
> > > > > >
> > > > > > If we look at Figure 3, most regions away from the training data have a very high fingerprint loss and these regions are mostly flat. The loss surface is essentially flat everywhere, except for a few valleys of low-loss around the data-distribution. We believe the loss surface for the fp-loss looks similar to this plot for MNIST and CIFAR-10 as well.  At large distances, for randomly sampled points, the gradient is essentially not providing much useful information. This interpretation is to some extent justified by what we observe for the random adversary (end of the appendix).
> > > > > >
> > > > > > If we interpret “gradient masking” as a phenomena of supplying the adversary with misleading gradients, we do not think NeuralFP does gradient masking. Also, optimization methods like SPSA that “locally” smoothen out the gradients may not yield much useful information on how to traverse the loss-surface at these “flat” regions to converge to something that fools NeuralFP.
> > > > > >
> > > > > > Some form of macro-scale loss-approximation or search techniques could perhaps be useful, and would advance the debate, leading to better attacks. We agree that evaluating the effects of the different hyper-parameters, particularly N would be a very interesting direction for future investigation.

---

### Public Comment · (anonymous) · 2018-11-05
**Suggestions**

Hi, I have some suggestions for the sanity check.

1) For all attacks, what happens if an adversary only considers \gamma L_{fp}? (i.e., an adversary does not make an input to be misclassified, but only want to make it a fake.)

2) How is the detection rate on Gaussian noise ~ N(0, \eta^2)?

---

> ### Author Response · Authors · 2018-11-05
> **Reply to Suggestions**
>
> Thanks for the suggestions for sanity checks. We will get back in more detail on this, but some initial comments are below:
>
> 1. During our initial tests we have tried this for FGSM. It turns out that the perturbations become nearly indistinguishable  (AUC-ROC ~~0.5-0.6) but the adversarial success rate drops to 0. We will try this for the other attacks (and FGSM again to confirm) and get back shortly.
>
> Attacking with only \gamma*L_{fp} is loosely equivalent to choosing a very large coefficient for L_fp (at which point the misclassification starts to fail).
>
> 2. We have tried detecting random images (every pixel drawn uniformly) and the AUC-ROC was ~~100%. We will get back on the detection of gaussian noise with varying eta shortly.

---

### Author Response · Authors · 2018-11-11
**Summary and general comments**

We thank all reviewers for their thoughtful comments and reviews. We would like to briefly summarize and highlight our contributions here, and address specifics in individual comments to the reviewers.

Some merits of our contribution we would like to highlight:

1. NFP is a blackbox defense, i.e. it is attack agnostic. To the best of our knowledge, this is the first such defense that does not use adversarial examples to learn to distinguish between real and adversarial images, but rather relies on the training data alone and still manages to achieve near perfect detection in the experiments described.

2. NFP does not require extensive fine-tuning to be effective. Indeed, our experiments show that NFP is effective against the strongest state-of-the-art attacks over a wide range of hyperparameters.

Despite the lack of formal guarantees, we present an abundant amount of empirical evidence that our method is robust against a full-range of the strongest available attacks in the literature. In sum, NFP serves as a strong baseline for developing stronger attacks. In turn, this can eventually lead to better defenses.

3. NFP is a promising step towards the problem of detecting and defending against  adversarial attacks with unbounded perturbations (we test SPSA, FGSM with eps=0.25 for CIFAR-10) -- particularly, when the fingerprint is not known to the adversary (for e.g., consider the challenge https://github.com/google/unrestricted-adversarial-examples).

---

### Public Comment · (anonymous) · 2018-11-22
**Clean VS Adversarial Examples**

How does the fingerprint work on clean images that are misclassified? Does it detect them as clean or adversarial?

---

> ### Author Response · Authors · 2018-11-23
> **Interesting Question!**
>
> We have seen that the detection AUC-ROC for a set of correctly classified vs adversarial examples is usually a bit higher than that for a set of clean (includes misclassified) vs adversarial.  Here, the correctly classified examples being from the clean set. This seems to suggest that the distribution of losses in misclassified samples is generally higher than that of correctly classified samples. Specifically how much higher is something we have not explicitly looked at, we can probably take a look at that after the rebuttal phase is complete.

---

### Author Response · Authors · 2018-11-26
**New experiments and revision**

We thank the reviewers again for the comments and insightful feedback. We report results on additional experiments to evaluate whether adversarial attacks with more iterations/random restarts can degrade the detection performance for NeuralFP. These experiments were run (for CIFAR-10 samples) following the feedback/suggestions provided in the reviews.

Attack                                                             Distortion        Iteration         Bisection       Step-size       AUC-ROC
                                                                           Bound                Steps              Steps                                     (%)
CarliniL2-FP (N=30, vareps=0.003)           Unbounded        20000                 20, 10              --                    94.12
CarliniL2-FP (N=30, vareps=0.003)           Unbounded        15000                 15,  9               --                    95.48
CarliniL2-FP (N=50, vareps=0.003)           Unbounded        20000                 20,  10             --                    95.56
Adaptive-PGD (l-2)                                            10.0             Steps = 100            6                    0.5                  99.37
                                                                                                  Restarts=5
Adaptive-PGD (l-2)                                            60.0             Steps = 150            6                    0.5                  99.55
                                                                                                  Restarts=5
Adaptive-PGD (l-inf)                                     16/255             Steps = 50              6                   0.005              99.21
                                                                                                  Restarts=50
Adaptive-PGD (l-inf)                                     16/255             Steps = 1000          6                   0.005              99.71
                                                                                                   Restarts=1
Adaptive-PGD (l-inf)                                      0.25                Steps = 1000           6                   0.005             99.48
                                                                                                   Restarts=1
Adaptive-PGD (l-inf)                                      1.0                  Steps = 150             6                    0.01               99.74
                                                                                                   Restarts=5
Adaptive-SPSA                                                 0.05               Steps = 1000           20                  lr=0.01          99.84
								            								               delta=0.01
Adaptive-SPSA                                                 0.05               Steps = 1000           20                  lr=0.005        99.71
								            								               delta=0.005
Adaptive-SPSA                                                 0.25              Steps = 1000           20                  lr=0.01           99.32
								            								            delta=0.01

1. To conclude, we see that state-of-the-art attacks with more iterations and varying hyperparameters search do *not* generate significantly more successful adversarial perturbations. Together with the results in our paper, this strongly suggests that current attacks are not able to effectively find adversarial perturbations that can fool NFP.

2. Moreover, generating adversarial perturbations by randomly sampling around the data is not successful. This is another sanity check that indicates that NFP does *not* merely implement gradient masking.

---

> ### Author Response · Authors · 2018-11-26
> **Contd.**
>
> We have included these results and discussion in the appendix in the revised submission.
>
> With that, we would like to emphasize a few points in addition to our earlier comments.
>
> ** Black-box/grey-box settings
> We believe NeuralFP is a strong defense in the black-box/grey-box settings. In these settings, NeuralFP achieves AUC-ROCs of 99-100% (near-perfect detection) consistently across attacks and in practical applications where security is a concern in the black-box setting, we believe NeuralFP is a significant step towards solving the problem of adversarial robustness.
>
> ** Adaptive white-box setting
> In the adaptive white-box setting, we show that the approach is robust over the full range of currently known attacks, even when run for a very large number of iterations, and with randomized restarts. We believe that running the attacks for much longer is not likely going to change the numbers significantly. (See Appendix H, Appendix I, Appendix J)
>
> ** Weakness of current attacks
> We do not claim that NeuralFP is entirely robust to all possible attacks, but against attacks that defenses are currently evaluated against, NeuralFP is able to detect such attacks quite successfully. These attacks fail even in scenarios when adversarial examples definitely exist, pointing to weaknesses in the current optimization methods.
>
> ** Gradient masking unlikely
> In addition to evaluation against black-box adversaries and adaptive white-box adversaries, we further evaluate against an adversary that functions by random sampling. This is suggested as a test for obfuscated gradients in (https://arxiv.org/pdf/1802.00420.pdf ). For a set of MNIST and CIFAR-10 unseen data-points, we sample large numbers of points (5e5 to 1e6 points for each input) uniformly randomly in epsilon-balls around the unseen data-points. For these points we analyse the fingerprint-losses and we find that for the entire set of points sampled, we cannot find a single adversarial point with a low fingerprint loss (See Appendix K).
>
> Further,  we  would  like to note that our defense does not  introduce any non-differentiable functions/optimization procedures that have frequently been associated with obfuscated gradients.
>
> ** Current attacks are not effective, even with more computation and hyperparameter search.
> Further, that current attacks fail even when allowed a large perturbation budget indicates the need for stronger attacks/improved optimization methods.
>
> In short, we believe that NeuralFP is a strong detection method for current attacks and serves as a useful baseline for developing a new range of adversarial attacks and thereby, robust defenses.

---

### Author Response · Authors · 2018-11-27
**Source Code and Model Weights at the following link**

https://www.dropbox.com/sh/iq0yub74gquz1od/AADZXUVabMvZasPrt-6-c-Wpa?dl=0

---

### Public Comment · (anonymous) · 2018-11-29
**Attack on training or test data?**

Are attacks applied to training or test data?

---

> ### Author Response · Authors · 2018-11-29
> **Unseen test data**
>
> The attacks are applied to unseen test data (mentioned in the experiments section 3.1).

---

### Public Comment · (anonymous) · 2018-12-10
**About the detection's performance**

I wonder that how is the performance of detection is there's no fingerprint embedded in your model bcz I think the detection method is useful enough. Looking forward to your reply.

---

> ### Author Response · Authors · 2018-12-10
> **Adv eg. vs clean indistinguishable**
>
> We have tried this earlier, and the AUC-ROC drops to ~~0.5. The adversarial examples are virtually indistinguishable from real ones if you do not embed the fingerprints.
>
> Does that answer your question? You could probably try to learn the "fingerprints" naturally embedded but I am not confident that the performance would be comparable.

---

> > ### Public Comment · (anonymous) · 2018-12-11
> > **efficiency of fingerprinting**
> >
> > Following up the previous question, there is a related paper: https://arxiv.org/abs/1709.05583
> > Basically, it can be seen from the paper that for adv, random perturbing will lead it back to the ground truth, while random labels for benign instances. So it is not clear how much value the fingerprinting adds here. It would be helpful to show some baseline results for without fingerprinting, and different numbers of adding the fingerprinting (the paper uses 5) to get better understanding. Thank you!

---

> > > ### Author Response · Authors · 2018-12-11
> > > **Effect of fingerprinting**
> > >
> > > I am assuming your question refers to the value of embedding during training. We can run the specific comparison you request.
> > >
> > > Also, I believe the paper uses m=1000 (number of "fingerprints") -- it does not use the change in normalized-logits, rather it considers the label predictions at a specific set of randomly sampled points around the data-point and uses that as an ensemble classifier. Here are some initial thoughts on the need for embedding:
> > >
> > > 1. We investigate the effect of adding different numbers of fingerprints in our paper. For instance, see Figure 9b: If we choose low fingerprint numbers  (say N=3) during embedding, the AUC-ROC is quite bad. In fact, we find that with an adaptive attack, we can make the adversarial examples seem more real than the real examples for such low fingerprint numbers.
> > >
> > > 2. The premise for https://arxiv.org/abs/1709.05583 is that adversarial examples are not robust, however there is some evidence that robust adversarial examples can be synthesized (See https://arxiv.org/abs/1707.07397).  We believe high-confidence adversarial examples are more robust (in relation to random samples in the neighborhood) => increased success rates for high confidence attacks in https://arxiv.org/abs/1709.05583. We believe this would not be the case if the fingerprints are embedded, we will run tests to confirm this.
> > >
> > > 3. Assuming a non-adaptive attacker, the success rates reported in the paper you cite (e.g. 78% adversarial success rate for JSMA on CIFAR-10) are significantly higher than those reported in our paper.
> > >
> > > 4.  https://arxiv.org/abs/1709.05583 uses randomized ensembles, so I suspect an adaptive attack with EOT (see Athayle et al. ICML'18) might succeed to some extent against the defense proposed in https://arxiv.org/abs/1709.05583.

---

### Public Comment · (anonymous) · 2018-12-13
**About targeted and untargeted**

I found that there's no mention about if these attack are targeted or untargeted but I guess they're all targeted. I wonder if there's some untargeted adv example detection experiments since I doubt that this method is not working well on untargeted comparing to targeted. Thanks!!!

---

> ### Author Response · Authors · 2018-12-13
> **Plenty of untargeted attacks in the paper**
>
> The SPSA, FGSM, PGD, BIM attacks are untargeted. This is already mentioned in the paper in the section on experiments. Please refer to the paper (end of page 7) and code for details.

---

### Meta-Review · Area_Chair1 · 2018-12-15
**interesting direction with extensive experiments but critical flaw**

**Confidence:** 2
**Recommendation:** Reject

**Metareview:**

* Strengths

The paper proposes a novel and interesting method for detecting adversarial examples, which has the advantage of being based on general “fingerprint statistics” of a model and is not restricted to any specific threat model (in contrast to much of the work in the area which is restricted to adversarial examples in some L_p norm ball). The writing is clear and the experiments are extensive.

* Weaknesses

The experiments are thorough. However, they contain a subtle but important flaw. During discussion it was revealed that the attacks used to evaluate the method fail to reduce accuracy even at large values of epsilon where there are simple adversarial attacks that should reduce the accuracy to zero. This casts doubt on whether the attacks at small values of epsilon really are providing a good measure of the method’s robustness.

* Discussion

There was substantial disagreement about the paper, with R1 feeling that the evaluation issues were serious enough to merit rejection and R3 feeling that they were not a large issue. In discussion with me, both R1 and R3 agreed that if an attack were demonstrated to break the method, that would be grounds for rejection. They also both agreed that there probably is an attack that breaks the method. A potential key difference is that R3 thinks this might be quite difficult to find and so merits publishing the paper to motivate stronger attacks.

I ultimately agree with R1 that the evaluation issues are indeed serious. One reason for this is that there is by now a long record of adversarial defense papers posting impressive numbers that are often invalidated within a short period (often less than a month or so) of the paper being published. The “Obfuscated Gradients” paper of Athalye, Carlini, and Wagner suggests several basic sanity checks to help avoid this. One of the sanity checks (which the present paper fails) is to test that attacks work when epsilon is large. This is not an arbitrary test but gets at a key issue---any given attack provides only an *upper bound* on the worst-case accuracy of a method. For instance, if an attack only brings the accuracy of a method down to 80% at epsilon=1 (when we know the true accuracy should be 0%), then at epsilon=0.01 we know that the measured accuracy of the attack comes 80% from the over-optimistic accuracy at epsilon=1 and at most 20% from the true accuracy at epsilon=0.01. If the measured accuracy at epsilon=1 is close to 100%, then accuracy at lower values of epsilon provides basically no information. This means that the experiments as currently performed give no information about the true accuracy of the method, which is a serious issue that the authors should address before the paper can be accepted.